# Contrast-enhanced ultrasound with sub-micron sized contrast agents detects insulitis in mouse models of type1 diabetes

David G. Ramirez[1], Eric Abenojar [2], Christopher Hernandez[3], David S. Lorberbaum[4], Lucine A. Papazian[4], Samantha Passman[1], Vinh Pham[1], Agata A. Exner [2,3✉] & Richard K. P. Benninger [1,4✉]

In type1 diabetes (T1D) autoreactive T-cells infiltrate the islets of Langerhans, depleting insulin-secreting β-cells (insulitis). Insulitis arises during an asymptomatic phase, prior to clinical diagnosis of T1D. Methods to diagnose insulitis and β-cell mass changes during this asymptomatic phase are limited, precluding early therapeutic intervention. During T1D the islet microvasculature increases permeability, allowing nanoparticles to access the micro-environment. Contrast enhanced ultrasound (CEUS) uses shell-stabilized gas bubbles to provide acoustic backscatter in vasculature. Here, we report that sub-micron sized 'nano-bubble' ultrasound contrast agents can be used to measure increased islet microvasculature permeability and indicate asymptomatic T1D. Through CEUS and histological analysis, pre-clinical models of T1D show accumulation of nanobubbles specifically within pancreatic islets, correlating with insulitis. Importantly, accumulation is detected early in disease progression and decreases with successful therapeutic intervention. Thus, sub-micron sized nanobubble ultrasound contrast agents provide a predicative marker for disease progression and therapeutic reversal early in asymptomatic T1D.

[1] Department of Bioengineering, University of Colorado Anschutz Medical Campus, Aurora, CO, USA. [2] Department of Radiology, Case Western Reserve University, Cleveland, OH, USA. [3] Department of Biomedical Engineering, Case Western Reserve University, Cleveland, OH, USA. [4] Barbara Davis Center for Diabetes, University of Colorado Anschutz Medical Campus, Aurora, CO, USA. ✉email: Agata.Exner@Case.edu; Richard.Benninger@CUAnschutz.edu

Type 1 diabetes (T1D) is caused by infiltration of auto-reactive T-cells into the pancreatic islets and the destruction of the insulin-producing β-cells[1]. Prior to the clinical presentation of diabetes there exists an asymptomatic phase where insulitis and immunological irregularities are present, but there is sufficient β-cell mass and insulin secretion to regulate blood glucose levels. After substantial β-cell loss (e.g. >80%[2]) patients present with hyperglycemia and are diagnosed with T1D[3]. The asymptomatic phase presents an opportunity for therapeutic intervention to blunt insulitis and preserve β-cell mass[4,5]. However, the inability to effectively diagnose insulitis and β-cell mass decline will limit attempts to treat patients prior to T1D onset. The presence of circulating auto-antibodies can predict eventual T1D onset, but these antibodies are not pathogenic and represent an aggregate risk of developing the disease[6]. Therefore, a method to identify and track the underlying disease progression during the asymptomatic phase is paramount for diagnosing and treating patients at risk for developing T1D.

To date, therapeutic interventions aimed at preventing islet autoimmunity and insulitis and preserving β-cell mass have had limited success. For example, the immunotherapy anti-CD3 (αCD3) has been shown to preserve insulin secretion (c-peptide levels) for up to two years. However, αCD3 treatment is highly heterogenous, where some patients retained robust insulin secretion more than two years ("responders"), but others lacked any preservation of insulin secretion ("non-responders")[7]. A means to assess the reversal of insulitis and preservation of β-cell mass would also enable the efficacy of preventative treatments to be assessed. However, there are limited means to monitor the response to preventative treatments.

There are challenges associated with imaging the state of the islet in vivo. The islets represent a low volume (1–2%) of the total pancreas mass[7]. As such any label for the islet can suffer from poor signal-to-background; for example the background can be ~50% greater than islet signal for GLP1R radio-labels[8]. The pancreatic islets do receive a disproportionately high proportion of the pancreas blood flow compared to the exocrine tissue (10–20%)[9,10] and islet blood flow can vary during the progression of insulitis and T1D[11–13]. Within the islet the microvasculature becomes more permeable during insulitis and inflammation, as demonstrated by the uptake of nanoparticles into the islet microenvironment of non-obese diabetic (NOD) mice[14] or STZ-treated mice[15], which model T1D. Magnetic resonance imaging (MRI) has been used to visualize magnetic nanoparticles (MNPs) uptake and retention within the inflamed islets during the development of T1D in rodent models[14–17] and in human T1D[18,19]. As such, measurements of increased islet-vascular permeability and changes in these measurements over time could be used more broadly to diagnose and monitor islet infiltration and decline during the progression of T1D.

Ultrasound is a cost-effective, easily deployable, and safe medical imaging modality. Contrast-enhanced ultrasound (CEUS) utilizes lipid- or protein stabilized gas core microbubbles (MBs) that provide a strong acoustic backscatter of ultrasonic waves. MBs are FDA approved for cardiac and liver imaging in adult and pediatric populations[20], and have been used "off label" for other indications[21,22]. While MBs are restricted to the blood stream, smaller sub-micron sized nanobubbles (NBs) are capable of extravasating through the hyper-permeable microvasculature of injured tissue. This includes accumulation of ~100–300 nm diameter NBs within tumors as a result of increased tumor vascular permeability[23–25]. Such NBs are echogenic at clinically relevant ultrasound frequencies (6–18 MHz), allowing regions of high permeability microvasculature to be imaged non-invasively and in real time.

In this study, we test whether CEUS measurements of sub-micron sized NBs can detect changes in islet microvascular permeability as a result of increasing insulitis during the asymptomatic phase of T1D. We find that NBs accumulated within the pancreas and islets in multiple mouse models of T1D compared to non-diabetic controls, with accumulation detectable at very early stages of disease. This accumulation can be measured by both ultrasound contrast signal and by histology, and correlates with the level of insulitis.

## Results

**Nanobubble contrast increases within the pancreas in T1D.** Previous work has demonstrated that sub-micron sized "nanobubbles" (NBs) accumulate within tumors that exhibit increased microvascular permeability. As such, we hypothesized that NBs would accumulate within the pancreatic islets undergoing insulitis where there is increased vascular permeability. Prior work examining vascular permeability using NBs and CEUS has shown an elevated signal immediately following injection corresponding to rapid filling of the vasculature, followed by a persisting signal for ~10 min corresponding to tissue accumulation. To examine these kinetics we first infused NBs via tail vein catheter into 10w NOD mice, at which age islets show heavy insulitis, and examined the kinetics of NB signal within the pancreas. We also performed these measurements utilizing different NB concentrations, center frequency and peak negative pressure (mechanical index, MI) in order to determine the conditions that most effectively detected the sub-micron sized NBs.

The pancreas tail can be readily located in B-mode ultrasound using the kidney, spleen and stomach as guide markers, as well by the distinct texture (Fig. 1a, Supplementary Fig. 1). Measurements in nonlinear contrast mode following NB infusion showed increased sub-harmonic contrast signal within ~1 min of infusion in the pancreas at 12.5 MHz center frequency (4% power), that returned to baseline around 5 min, likely corresponding to vascular filling (Fig. 1b)[11]. This immediate elevation in contrast signal was diminished when using a center frequency of 18 MHz, which is closer to the resonance frequency of smaller sized bubbles. With a center frequency of 18 MHz and higher transmission power (10%), an elevation in contrast signal that was maintained over ~30 min was observed in the pancreas (Fig. 1c). Continual imaging for >30 min using 18 MHz and 10% transmission power did not significantly impact the contrast signal generated by pancreas-accumulating NBs (Supplementary Fig. 2a). No elevation in contrast was observed with a center frequency of 12.5 MHz at a higher transmission power, even with intermittent imaging (Supplementary Fig. 2b). Thus, a sustained increase in contrast signal is observed within the pancreas associated with NB infusion.

Given increased islet microvascular permeability during the progression of T1D, we next sought to determine if NB signal within the pancreas was associated with disease. In 10w NOD mice NB contrast in the kidney, where no immune infiltration occurs, did not increase significantly after ~30 min, and was significantly less than the substantial elevation in contrasts signal within the pancreas (Fig. 1d,e). Following saline vehicle infusion both the kidney and pancreas did not show a signal increase (Fig. 1f), indicating signal specifically associated with the infused NB formulation. To further examine the disease-dependence of NB signal changes within the pancreas, we compared immunodeficient NOD;Rag1ko (Rag1ko) mice which do not develop spontaneous diabetes. After NB infusion, there was no significant increase in NB contrast signal in the pancreas of Rag1ko mice (Fig. 1g) compared to the robust elevation in NOD mice. In

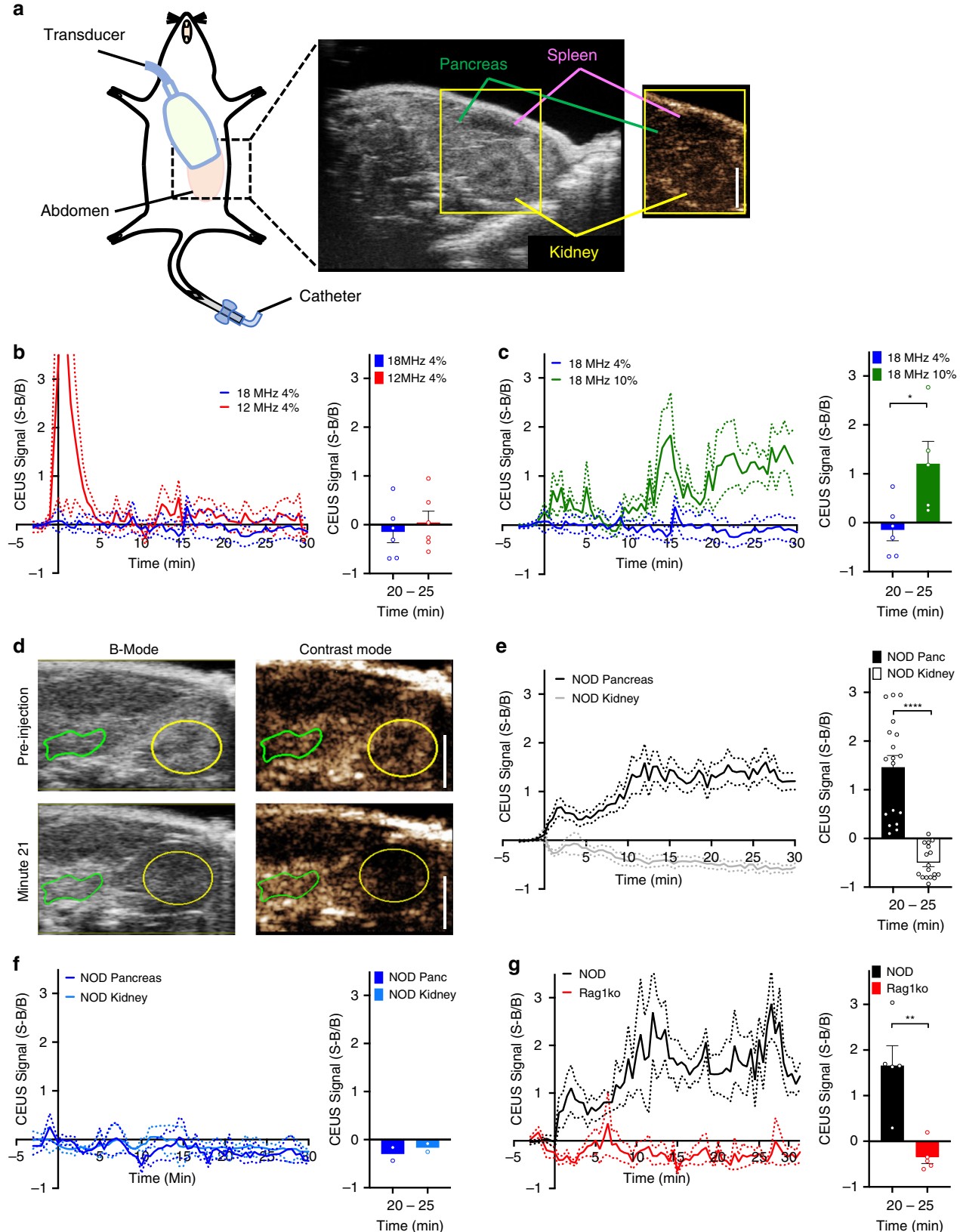

healthy 10w female C57BL/6 mice, there also was a lack of any significant signal change following NB infusion (Supplementary Fig. 3).

These results show that there is increased NB contrast signal specifically within the pancreas in the NOD mouse models of T1D, where increased islet microvascular permeability occurs. The signal does not increase in the kidney nor in immune deficient Rag1ko mice and healthy C57BL/6 mice. This is consistent with the changes in islet microvascular permeability induced by insulitis prior to T1D onset.

**Fig. 1 Contrast-Enhanced Ultrasound measurements of NB targeting the pancreas in T1D. a** Schematic illustrating the transducer placement on the animal models (left) and the anatomical landmarks used to identify the pancreas during the ultrasound scan (right). The pancreas (green), spleen (purple), and kidney (yellow) are shown in B-Mode and Contrast Mode. **b** Mean time-course of contrast signal in the pancreas (Panc) of 10-week-old female NOD mice following NB infusion, using 12.5 MHz or 18 MHz center frequency, each at 4% transmission power (left), together with the mean contrast signal averaged between 20 and 25 min (right). **c** Mean time-course of contrast signal in the pancreas of 10-week-old female NOD mice following NB infusion, using 18 MHz center frequency, at 4% or 10% transmission power (left), together with the mean contrast signal averaged between 20 and 25 min (right). **d** Representative B-mode and sub-harmonic contrast ultrasound images of pancreas (green) and kidney (yellow), before and after NB infusion, at 18 MHz and 10% power. **e** Mean time-course of contrast signal in the pancreas and kidney of 10-week-old female NOD mice following NB infusion (left), together with the mean contrast signal averaged between 20 and 25 min (right). **f** Mean time-course of contrast signal, and mean signal between 20 and 25 min as in (**e**) comparing pancreas and kidney of 10-week-old female NOD mice following saline infusion. **g** Mean time-course of contrast signal, and mean signal between 20 and 25 min as in (**e**) comparing pancreas of 10-week-old female NOD mice and 10-week-old female NOD;Rag1ko (Rag1ko) mice following NB infusion. Dashed lines in (**b**–**g**) represent 95% CI, error bars in (**b**–**g**) represent s.e.m. Data in (**b**, **c**) representative of $n = 6$ mice. Data in (**e**) representative of $n = 18$ mice (17 kidney measurements). Data in (**f**) representative of $n = 2$ mice. Data in (**g**) representative of $n = 6$ mice. Scale bar in (**a**, **d**) represents 3 mm. *$p < 0.05$, **$p < 0.01$, ***$p < 0.001$, ****$p < 0.0001$ comparing groups indicated (Paired Student's $t$ test, 2-sided). (**c**) $p = 0.0197$, (**d**) $p < 0.0001$, (**g**) $p = 0.0024$. Source data are provided as a Source Data file.

**Nanobubble contrast signal predicts disease progression**. Given elevated contrast in the pancreas of NOD mice following NB infusion, we next examined whether this contrast elevation could predict diabetes development. The time-course of contrast elevation can be quantified in several ways, including the time after infusion at which contrast increases, and the elevation of contrast at different time points (10–15 min and 20–25 min) after infusion (Supplementary Fig. 4a). We grouped the NOD mice by quartiles for each parameter. We then assessed the diabetes development for each quartile grouping. The time for contrast elevation and the mean contrast elevation at 10–15 min after infusion did not exhibit any obvious trend for diabetes development (Supplementary Fig. 4b, c). However, mean contrast elevation at 20–25 min after infusion did exhibit a trend, where NOD mice that showed low contrast elevation developed diabetes more slowly (Supplementary Fig. 4d). We next set thresholds to classify NOD mice as disease positive (high contrast elevation) or disease negative (low contrast elevation), based upon the contrast elevation measured in Rag1ko mice (Supplementary Fig. 4e). Those "disease negative" NOD mice that showed a low contrast elevation below the threshold, developed diabetes more slowly with reduced incidence (Supplementary Fig. 4f). For the lower of the thresholds chosen, this delay in diabetes development was statistically significant. Thus, contrast elevation measured following NB infusion is predictive of diabetes development.

**Nanobubble accumulation restricted to pancreatic islets**. To characterize the biodistribution of NBs within the pancreas in the presence of insulitis, we infused rhodamine-labeled NBs into 10w NOD and Rag1ko mice. Following infusion of these labeled NBs the contrast signal within the pancreas and kidney was similar to that following infusion of unlabeled NBs: only in the pancreas of the NOD mice was there a robust contrast signal increase (Fig. 2a, b). In NOD and Rag1ko mice we performed histological analysis of NB-rhodamine accumulation within the islets and exocrine tissues of the pancreas, where islets were identified by their characteristic autofluorescence and DAPI staining (Fig. 2c,d). In 10w NOD mice, islet regions of the pancreas showed significantly greater fluorescence coverage compared to exocrine regions (Fig. 2c,e), representing greater NB accumulation. Furthermore, islet regions from 10w NOD mice had significantly greater fluorescence coverage compared to islet regions from 10w Rag1ko mice (Fig. 2c,d,e). In NOD mice over 80% of the islets had some level of rhodamine coverage, representing NB targeting, and this was much lower in Rag1ko mice (Fig. 2f). The exocrine tissue from both models showed significantly less coverage than in 10w NOD islets (Fig. 2e). Thus, NBs target the islet regions preferentially compared to

exocrine regions in NOD mice but not immunodeficient mice that do not develop diabetes.

Prior studies have indicated nanoparticles can extravasate into the islet microenvironment during insulitis as a result of increased microvascular permeability[14,15]. To confirm NBs were extravasating from the microvasculature, as opposed to adhering to vessel walls, we performed histological analysis following infusion of rhodamine-labeled NBs and fluorescently conjugated tomato lectin to label blood vessels (Fig. 2g). Within the islets, we observed NB-rhodamine coverage was significantly less within the vasculature regions (lectin+) compared to outside the microvasculature regions (lectin-) (Fig. 2h), indicating extravasation of NBs into the islet microenvironment was occurring.

To test whether NB accumulation depended on the level of insulitis and thus disease, we scored the level of insulitis in 10w NOD and Rag1ko immunodeficient mice (Fig. 2i). We utilized adjacent sections stained with either hematoxylin and eosin (H&E) or measuring rhodamine coverage, and correlated the insulitis scoring and the fluorescent coverage in the islet region of both mouse models. 10w NOD mice showed substantial insulitis (peri-insulitis to infiltrating), with negligible levels in NOD; Rag1ko mice (Fig. 2j). The mean rhodamine coverage across islets of the pancreas of 10w NOD mice was significantly correlated with the mean insulitis score across islets of the same pancreas (Fig. 2k). Thus, mice with higher immune cell infiltration within the islets show greater NB targeting to the islets across the pancreas. However, when examined on an islet-by-islet basis, interestingly there was no significant relationship between islet insulitis score and fluorescent coverage (Fig. 2l). Thus, NB targeting of infiltrated islets is based on the pancreas-wide level of insulitis within the NOD mouse.

To test whether NB accumulation also correlated with β-cell decline we performed immunofluorescence for insulin and glucagon on sections from the same pancreas as rhodamine coverage was examined (Fig. 2m). NOD mice showed reduced insulin positive area compared to NOD;Rag1ko mice (Fig. 2n). There was a trend to declining insulin positive area with increased rhodamine coverage, but this was not statistically significant (Fig. 2o).

Taken together, these data suggest that increased islet microvascular permeability occurs alongside islet immune cells infiltration and, to a lesser degree, β-cell decline. Thus, we can observe increased insulitis by using extravasating NBs and ultrasound contrast signal measurements, as a measure for diabetes progression.

**Nanobubble accumulation within the islets is size-dependent**. The NBs are fractionated from a polydisperse sample that will also contain micron sized bubbles (microbubbles, MBs). Given

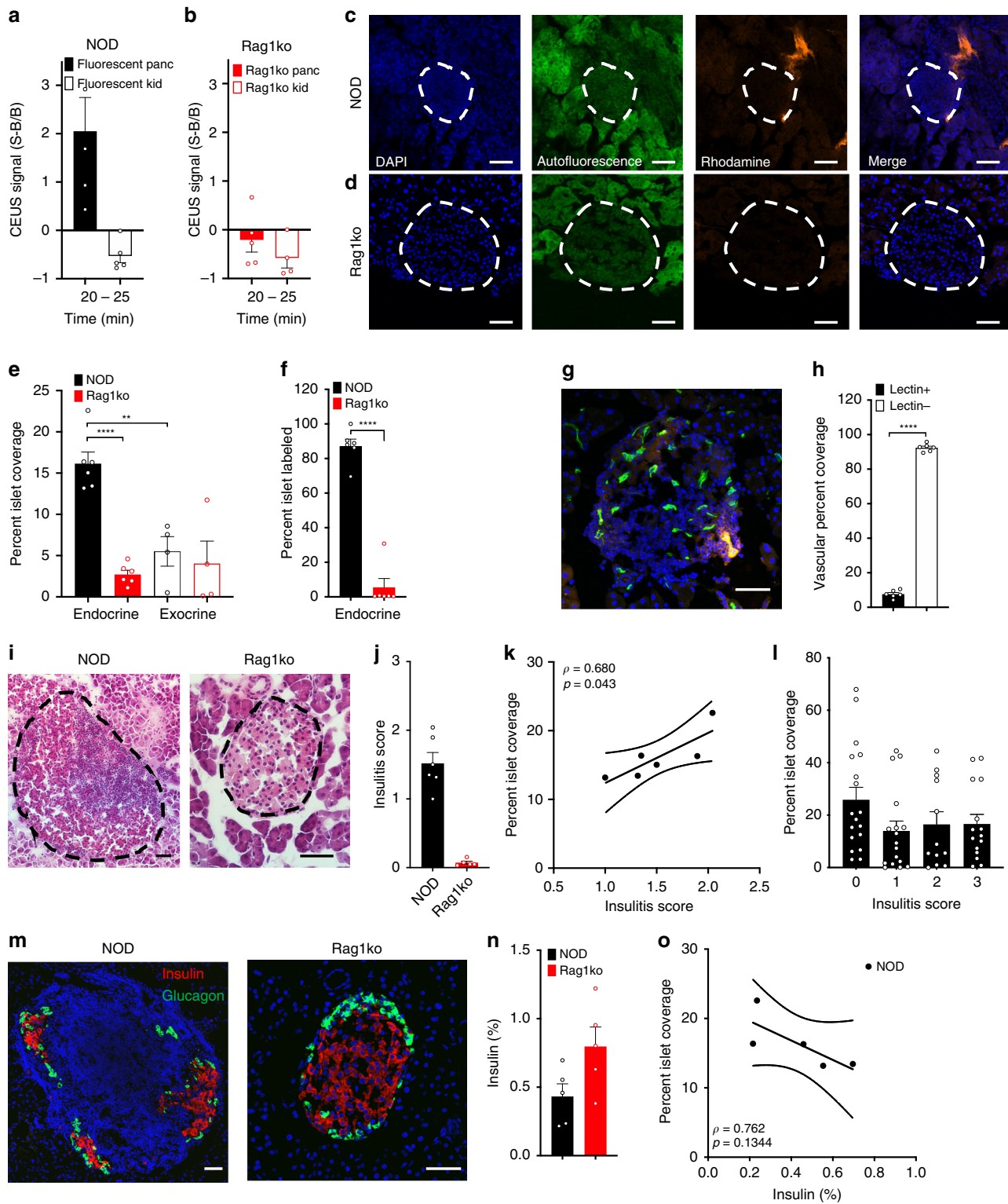

that the MB scattering cross-section scales strongly with bubble radius ($\sim r^6$)[26], even a very small population of MBs could explain the contrast signal increase within the pancreas. To determine whether sub-micron sized bubbles are required to generate the contrast within the pancreas we observe, we first characterized the sizes of the sub-micron NB fraction we infuse. The NB fraction contained approximately four orders of magnitude more "nano-sized" (<1 μm diameter) objects than "micron-sized" (>1 μm diameter) objects (Fig. 3a, b), as measured by microscopy

analysis. Similar results were obtained using Resonant Mass Measurement (RMM, Archimedes) (Supplementary Fig. 5), where the '"nano-sized" (196 ± 32 nm diameter) objects were also approximately four orders of magnitude more abundant than "micron-sized" (2356 ± 483 nm diameter) objects. The micron-sized MB fraction was substantially depleted of "nano-sized" objects, but contained a similar number of "micron-sized" objects as in the NB fraction (Fig. 3c,d). We infused 10w NOD mice with these different fractions on alternating days to test if the MB

**Fig. 2 Histological assessment of NB targeting the islets in T1D. a** Mean contrast signal between 20 and 25 min following Rhodamine-labeled NB infusion, in the pancreas (Panc) and kidney (Kid) of 10-week-old female NOD mice. **b** As in (**a**) for 10 week female Rag1ko mice. **c** Representative confocal images of an islet within a pancreas section of 10-week female NOD mouse following rhodamine-labeled NB infusion (orange). Islet is circled with a dotted line, as determined from autofluorescence (green) and DAPI-labeling morphology (blue). **d** As in (**c**) for 10-week-old Rag1ko female mice. **e** Mean rhodamine coverage in islet (Endocrine) and acinar (Exocrine) tissue in 10-week-old NOD female mice and 10-week-old Rag1ko female mice. **f** Mean proportion of islets with rhodamine labeling in 10-week-old NOD and Rag1ko mice. **g** Representative maximum-projection confocal image of an islet within a pancreas section of a 10w female NOD mouse infused with rhodamine-NBs (orange) and DyLight 488 tomato lectin (green). Islet circled with a dotted line, as determined from brightfield and DAPI-labeling morphology. **h** Mean rhodamine coverage within vascular areas (lectin positive) and non-vascular areas (lectin negative) in 10w NOD female mice. **i** Representative images of hematoxylin and eosin (H&E) stained pancreas sections of 10w NOD mice (left) and Rag1ko mice (right). Islet circled with a dotted line, as determined from brightfield morphology. **j** Mean insulitis score comparing 10w NOD and Rag1ko mice. **k** Scatterplot of NB-rhodamine fluorescent coverage of islets within the pancreas against the mean insulitis score within the pancreas, for NOD mice. **l** Mean NB-rhodamine fluorescent coverage of islets within the pancreas that show insulitis scores of 0, 1, 2, or 3 (see methods). **m** Representative images of insulin (red) and glucagon (green) and in pancreas sections of 10w NOD mice (left) and Rag1ko mice (right). **n** Mean insulin+ area as a fraction of total pancreas area. **o** Scatterplot of NB-rhodamine fluorescent coverage of islets within the pancreas against the mean insulin+ area, for NOD mice. Error bars in (**a, b, e, f, j, l, n**) represent s.e.m. Trend line in (**k, o**) indicates linear regression with 95% confidence intervals. Data in (**a**) represent n = 5 mice, (**b**) represents n = 5 mice (4 kidney measurements). Data in (**e, f**) represents 284 islets and 34 exocrine regions from n = 6 NOD mice and n = 6 Rag1ko mice. Data in (**h**) represent 66 islets from n = 3 mice. Data in (**j, k**) represent n = 6 NOD mice and n = 6 Rag1ko mice. Data in (**l**) represents 62 islets from n = 5 NOD mice. Data in (**n, o**) represent n = 5 NOD mice and n = 5 Rag1ko mice. Scale bar in (**c, d, g, i, m**) represents 50 μm. *p < 0.05, **p < 0.01, ***p < 0.001, ****p < 0.0001 comparing groups indicated (paired Student's t test, 2-sided, for data in (**a, b, f, h, n**); ANOVA for data in (**e, l**); Pearson's correlation for data in (**k**)). (**e**) p < 0.0001 (NOD vs Rag1ko endocrine), (**e**) p = 0.0015 (NOD exocrine vs endocrine), (**f**) p < 0.0001, (**h**) p < 0.0001. A mixed-effects model was used to assess the statistical significance and generate the regression in (**k,o**). Source data are provided as a Source Data file.

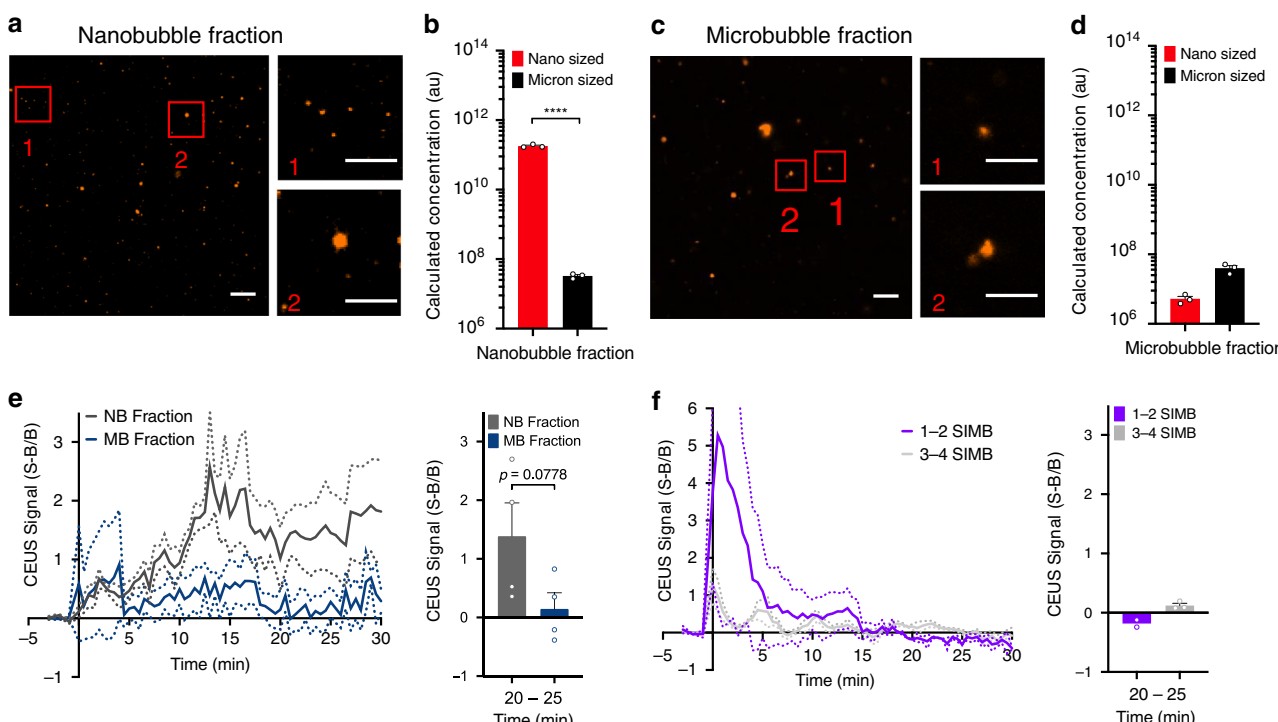

**Fig. 3 Size-dependence to NB targeting the pancreas in T1D. a** Representative images of nano-sized (label 1) and micron-sized (label 2) objects within the nanobubble fraction. **b** Calculation of the number of nano-sized and micron-sized objects within the nanobubble (NB) fraction. **c** As in (**a**) for the microbubble (MB) fraction. **d** As in (**b**) for the microbubble fraction. **e** Mean time-course of contrast signal in the pancreas of 10-week-old female NOD mice (left), together with the mean contrast signal between 20 and 25 min (right) following NB infusion (NB Fraction) or MB infusion (MB Fraction). **f** Mean time-course of contrast signal in the pancreas of 10-week-old female NOD mice (left), together with the mean contrast signal between 20 and 25 min (right) following infusion of size-isolated MBs of diameter 1–2 μm (SIMB 1–2) or 3–4 μm (SIMB 3–4). Dashed lines in (**e, f**) represent 95% CI, error bars in (**b, d, e, f**) represent s.e.m. Data in (**a–d**) represent n = 3 samples, with each sample assessed under three separate dilutions. Data in (**e**) represent n = 4 NOD mice. Data in (**f**) represent n = 3 NOD mice. Scale bars in (**a, c**) represent 10μm (large image) and 6 μm (small closeup images). *p < 0.05, **p < 0.01, ***p < 0.001, ****p < 0.0001 comparing groups indicated (paired Student's t test, 2-sided). (**b**) p < 0.0001. Source data are provided as a Source Data file.

fraction contributed to the pancreas contrast signal. While the NB fraction shows a sustained increase in contrast signal in the pancreas over 30 min, the MB fraction within the pancreas did not significantly differ from background (Fig. 3e). We also

infused ~10 million size-isolated microbubbles (SIMBs) of either 3–4 μm diameter or 1–2 μm diameter to test whether specific sized MBs could generate disease-dependent contrast signal within the pancreas. SIMBs of each size range showed a strong

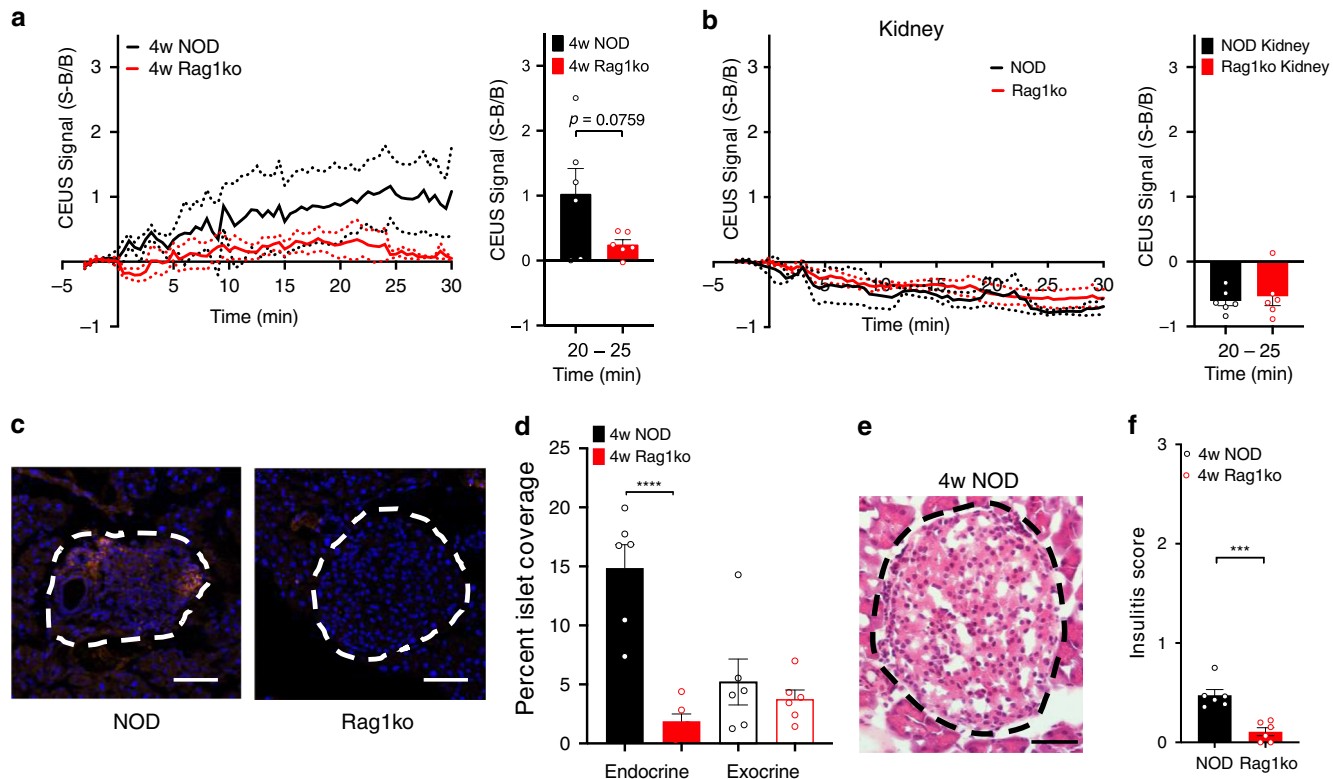

**Fig. 4 NBs target the pancreas and islet early in disease development. a** Mean time-course of sub-harmonic contrast signal in the pancreas of 4-week-old female NOD mice and 4-week-old female Rag1ko mice (left), together with the mean contrast signal averaged between 20 and 25 min (right). **b** As for (**a**) in the kidney of 4-week-old female NOD mice and 4-week-old female Rag1ko mice. **c** Representative images of an islet within the pancreas of a 4-week-female NOD and a 4-week-female Rag1ko mouse following rhodamine-NB infusion. **d** Mean rhodamine coverage in islet (Endocrine) and acinar (Exocrine) tissue in 4-week-old NOD female mice and 4-week-old Rag1ko female mice. **e** Representative images of hematoxylin and eosin (H&E) stained pancreas sections of 4w NOD mice. Islet circled with a dotted line, as determined from brightfield morphology. **f** Mean insulitis score comparing 4w NOD and Rag1ko mice. Error bars in (**a**, **b**, **d**, **f**) represent s.e.m. Data in (**a–f**) represent $n = 6$ NOD mice and $n = 6$ Rag1ko mice. Scale bar in (**c**, **e**) represents 50 μm. *$p < 0.05$, **$p < 0.01$, ***$p < 0.001$, ****$p < 0.0001$, comparing groups indicated (paired Student's $t$ test, 2-sided). (**d**) $p < 0.0001$, (**f**) $p = 0.0004$. Source data are provided as a Source Data file.

contrast signal within ~1 min, consistent with their high echogenicity and vascular filling. However, no signal increase was observed within the pancreas at >10 min after infusion (Fig. 3f), although 1–2 μm SIMBs cleared slightly slower than 3–4 μm SIMBs.

The lack of any pancreas contrast signal elevation from micronsized bubbles therefore indicates that NB pancreas contrast signal elevation and NB accumulation within the islets is dependent on the NB size.

**Nanobubbles target the islets in NOD mice at 4w.** To determine how early in the disease progression NB signal can indicate changes in the islet microvasculature, we examined 4w NOD and 4w Rag1ko mice. Following NB infusion, a sustained increase in contrast signal was observed in the pancreas of 4w NOD mice (Fig. 4a). This contrast signal was significantly greater than the contrast signal in the pancreas of 4w Rag1ko mice, which remined similar to background levels (Fig. 4a). However, the contrast signal in 4w NOD mice was less than the contrast signal we observed in 10w NOD mice (Fig. 1e, g). In both 4w NOD and Rag1ko mice there was no contrast elevation within the kidney (Fig. 4b). We also performed histological analysis on these same mice to quantify NB-rhodamine fluorescence coverage (Fig. 4c). The islet regions of 4w NOD mice had significantly increased rhodamine coverage compared to the islet regions of 4w Rag1ko mice (Fig. 4d). However, again the exocrine regions from both models showed similar coverage, which was significantly less than in 4w NOD islets. We observed some insulitis in islets of 4w NOD

mice, with negligible levels in 4w Rag1ko mice (Fig. 4e, f), indicating there was sufficient disease present in 4w NOD mice to cause nanobubble accumulation in the islets. These data suggest that NBs can accumulate within the pancreas alongside mild insulitis that occurs very early in the progression of T1D.

**Predicting therapeutically delayed diabetes.** Our histological data (Fig. 2) suggest that NB accumulation occurs alongside insulitis across the pancreas. To further test whether NB accumulation is related to insulitis and is not a specific feature of the NOD mouse, we utilized an adoptive transfer (AT) mouse model in which immunodeficient NOD-scid mice receive a transfer of diabetogenic splenocytes from recently diabetic NOD mice (Fig. 5a). In this AT model, mice develop diabetes between 5 and 8 weeks after splenocytre transfer (Fig. 5b). Following NB infusion, AT mice and control mice lacking an adoptive transfer showed similar contrast signal measurements at baseline prior to transfer (Fig. 5c). At 4 weeks post-transfer the AT mice had significantly greater contrast signal within the pancreas compared to control mice (Fig. 5d). Therefore, NB contrast signal can detect increased insulitis across the pancreas across a different models of T1D, and therefore is a reporter of insulitis and concurrent inflammation.

To determine whether measurement of NB accumulation by contrast-enhance ultrasound can report on therapeutic diabetes reversal we treated AT mice at 2 weeks post-transfer with antiCD4 antibody to deplete CD4[+] T cells[11]. AntiCD4 treatment significantly delayed diabetes by ~3 weeks (Fig. 5e). AntiCD4

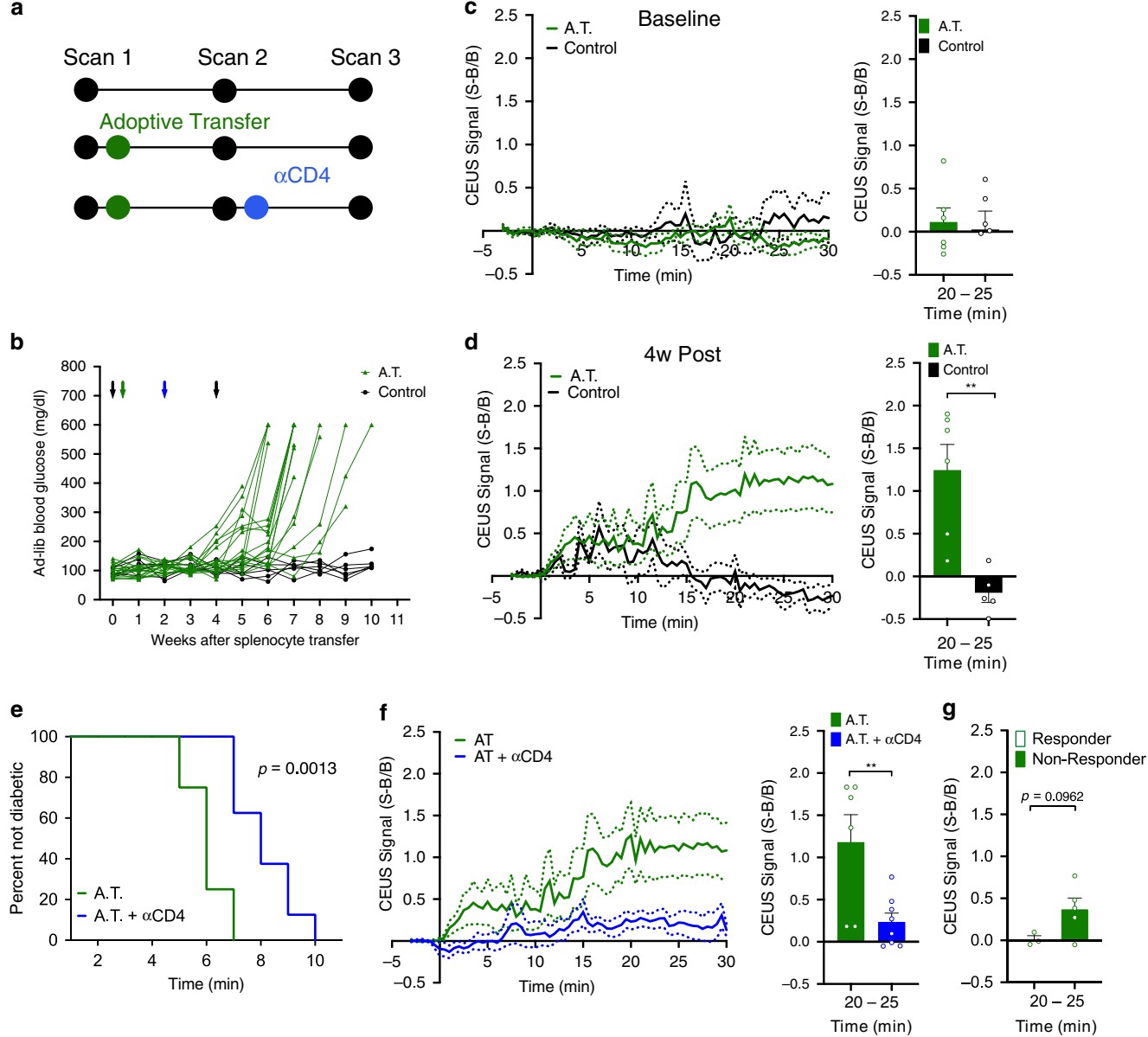

**Fig. 5 NBs targeting the pancreas is dependent on immune cell infiltration. a** Schematic of treatments and scan times for the adoptive transfer mouse model. **b** Time course of ad-lib blood glucose in the adoptive transfer (AT) model treated with splenocytes from diabetic female NOD donors (AT, green) or treated with vehicle alone (control, black). Black arrows indicate time points of contrast enhanced ultrasound scans, green arrow indicates delivery of splenocytes/vehicle. **c** Mean time-course of sub-harmonic contrast signal in the pancreas of female NOD-scid mice prior to splenocyte/vehicle delivery (left) together with the mean contrast signal between 20 and 25 min (right). **d** As in (**c**) for the pancreas of NOD-scid female mice 4w post splenocyte transfer. **e** Kaplan–Meier curve indicating proportion of mice remaining non-diabetic for AT mice and AT mice treated with antiCD4 at 2 weeks after transfer. **f** Mean time-course of sub-harmonic contrast signal in the pancreas of female NOD-scid mice 4w post splenocyte transfer (AT) and 4w post-transfer where antiCD4 is provided at 2w (left), together with the mean contrast signal between 20 and 25 min (right). **g** Mean contrast signal between 20 and 25 min for AT mice treated with antiCD4 that develop diabetes >8 weeks after splenocyte transfer ("Responder") or ≤8 weeks ("Non-Responder"). Error bars in (**c**, **d**, **f**, **g**) represent s.e.m. Data in (**b**) represent $n = 16$ NOD-scid mice. Data in (**c**, **d**) represent $n = 6$ A.T. mice and $n = 6$ Control mice. Data in (**e**–**g**) represent $n = 6$ AT mice and $n = 8$ AT + antiCD4 mice. *$p < 0.05$, **$p < 0.01$, ***$p < 0.001$, ****$p < 0.0001$, comparing groups indicated (Student's $t$ test, 2-sided). (**d**) (bars) $p = 0.0025$, (**f**) (bars) $p = 0.0087$. Source data are provided as a Source Data file.

treated AT mice showed significantly reduced contrast signal within the pancreas compared to untreated AT mice (Fig. 5f). Those antiCD4 treated AT mice that were substantially delayed in developing diabetes (>8 weeks after transfer, "responder") did not show a significant contrast elevation, whereas those antiCD4 treated AT mice that were not delayed in developing diabetes (≤8 weeks after transfer, "non-responder") still showed significant contrast elevation (Fig. 5g). Therefore, NB contrast signal can

detect and predict a therapeutically induced delay in diabetes development.

## Discussion

There are limited means for detecting the asymptomatic phase of type1 diabetes (T1D) development. Accurately detecting the progression of immune infiltration and islet decline would allow diagnosis of diabetes development to enable early preventative

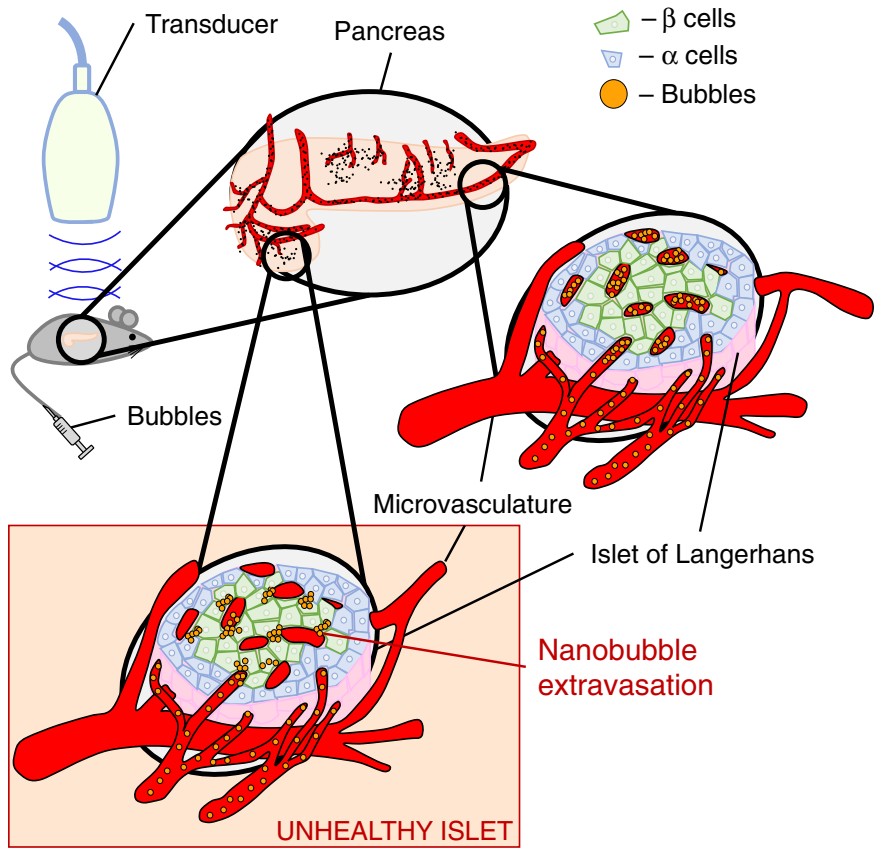

**Fig. 6 Summary of experiments demonstrating nanobubbles extravasate in diseased islets which increases the contrast signal within the pancreas.**
After NBs are delivered, the healthy islet microvasculature are not permeable to NB extravasation so the NBs are retained in the vasculature and cleared. Infiltrated islets in T1D have high vascular permeability and therefore NBs accumulate and are retained in the tissue following infusion, providing a source of sub-harmonic contrast. This sub-harmonic contrast signal therefore provides a diagnostic measure of insulitis and pre-clinical asymptomatic T1D.

treatment, as well being able to monitor the efficacy of this treatment. Here, our goal was to test whether changes in islet microvascular permeability that occurs concurrently with insulitis, local inflammation and islet decline could be detected using standard nonlinear ultrasound available on most clinical scanners and sub-micron sized nanobubble (NB) ultrasound contrast agents. We demonstrated that nanobubbles (NBs) accumulate specifically within the pancreas in multiple animal models of T1D as detected by persistent increased ultrasound sub-harmonic contrast, and that histological measurements further localize the accumulation to the infiltrated pancreatic islets. These results suggest that contrast-enhanced ultrasound using NB contrast agents can be used to detect and longitudinally track pre-symptomatic T1D progression (Fig. 6).

**Nanobubble CEUS detects insulitis and asymptomatic T1D.** Our results show strong evidence that CEUS with sub-micron sized NBs can detect insulitis and islet inflammation associated with the pre-clinical asymptomatic (pre-symptomatic) stage of T1D. This includes significant targeting to the pancreas, but not other imaged organs, as measured by ultrasound (Fig. 1); as well as specific targeting to the islets but not exocrine tissue as measured by histology (Fig. 2). The lack of pancreas and islet targeting in immune deficient mice that do not get diabetes further indicates this targeting is specific to models of type1 diabetes (Figs. 1,2). We observed similar results in both NOD animals (Fig. 1) and the adoptive transfer model (Fig. 5) in which insulitis leads to islet decline and diabetes. This indicates that islet targeting and accumulation is not model-specific, but occurs

alongside insulitis. The correlation between insulitis across the pancreas and nanobubble accumulation within the islets further indicates an ability to detect insulitis, although we cannot be sure that the accumulation is directly as a result of insulitis itself (see below). Nevertheless, non-invasive measurement of sub-micron sized bubbles accumulation within the islets using contrast-enhance ultrasound provides a convenient measure of the progression of insulitis and pre-symptomatic type1 diabetes.

We also observed that nanobubble accumulation coincided with reduced β-cell mass; when comparing NOD vs Rag1ko models and observing a trend across NOD animals (Fig. 2). However, this link was less strong than the link between nanobubble accumulation and insulitis. As such, the increased targeting that coincides with reduced β-cell mass is likely secondary to increased insulitis or some other factor associated with diabetes progression.

We observed nanobubble accumulation and contrast signal within the pancreas in 10w NOD mice. However, we also observed significant ultrasound contrast within the pancreas and nanobubble accumulation within the islets in 4w NOD mice, albeit lower than measured in 10w NOD mice. At this age insulitis is very mild, consisting mostly of peri-insulitis[13,27–29], yet nanobubble accumulation was readily detected. Thus, measurement of nanobubble accumulation is very sensitive to disease development. This finding also compares favorably to other modalities that have been applied to pre-clinical models. For example, insulin auto-antibodies are elevated after 4 weeks in NOD mice[30]. Magnetic nanoparticles show significant accumulation within the islets as early as 6 weeks in NOD mice or 4 weeks in NOD-BDC2.5 mice[14]. Notably, clear changes were observed with measurements at a single time point:

multiple longitudinal measurements may capture the progression of insulitis more clearly. Nevertheless, such highly sensitive accumulation and thus measurement of insulitis may have drawbacks. For example, predicting diabetes development was most effective for those animals that develop diabetes very late or not at all, and was less effective at identifying very rapid progression to diabetes (Supplementary Fig. 4). Nevertheless, we were able to detect a delay in disease upon therapeutic intervention. Testing whether effective disease reversal, induced by other therapies, is predictable by nanobubble accumulation measurements will be a key goal for future work.

Insulitis and inflammation contribute to the increased NB signal within the pancreas and islets, likely as a result of increased microvascular permeability within the islet, Interestingly while the level of NB accumulation across the pancreas correlated with the level of insulitis across the pancreas, there was no such correlation by islet: islets with substantial infiltration on average showed similar NB targeting as islets showing peri-insulitis or absent insulitis within the same pancreas. Such islet-by-islet analysis has not been presented within the literature and it is unclear what may be causing NB targeting in low infiltrated islets. For example inflammatory mediators may act across all islets irrespective of the level of insulitis[31]. As such, NB accumulation reflects the level of infiltration and inflammation across the pancreas. While islet microvascular changes occur in mouse models of T1D and human T1D[6,16,18], it has not been demonstrated exactly what causes these microvascular changes and whether they solely reflect the islet microenvironment or are influenced by events elsewhere in the pancreas. For example, sympathetic activity is a strong tonic regulator of islet pericyte function[32]. Determining the molecular mechanism underlying these microvascular changes that reflect NB accumulation, and likely the accumulation of other imaging contrast agents, is therefore needed.

Our measurements indicate that nanobubbles specifically accumulate within the islets in models of T1D. Nanobubbles can be readily functionalized with antibodies and peptides, and can also incorporate hydrophobic small molecules. This property has been utilized for both molecular imaging[25] and also for specific therapeutic delivery[33]. While microbubbles have been used for cargo delivery[34,35], their localization is restricted to the vasculature and requires ultrasound-induced "ablation" for cargo release. Given the passive delivery of nanobubbles to the islet, they may also provide an effective vehicle for therapeutic delivery to this region.

**Increased contrast signal correlates to islet-vascular changes.** Our results show a clear correlation between NB accumulation and insulitis. The islet-vascular interface has many fenestrae to facilitate the transfer of cellular products, with the microvasculature within the islet normally being permeable to particles of <70 kDa, equivalent to ~6 nm size[36]. This is much greater than the microvasculature within the exocrine tissue which is permeable to particles of <20 kDa, equivalent to 2–3 nm size[36]. However, under inflammatory conditions in T1D the islet microvascular permeability specifically increases, allowing passage of larger molecules, for example ~25 nm magnetic nanoparticles[14–16]. The NBs we utilized have a size of 100–300 nm diameter which is consistent with the object size that is permeable in models of T1D. We also demonstrated that bubbles of >1 μm diameter did not extravasate and accumulate, which is also consistent with our prior measurements of microbubbles within the pancreas in models of T1D[11], as well as being consistent with use of high MW fluorophore dextrans for labeling the vasculature[36,37]. These results are also consistent with prior imaging of NB accumulation within tumors by ultrasound and histology[33,38,39].

Our results support that extravasation through more permeable islet microvasculature and accumulation within the parenchyma is occurring in models of T1D. As such targeting of nanobubbles is to the region of the islet defined by the extent of increased microvasculature permeability, and not specifically to β-cells. However, it is important to note that we observed a very small component of micron-sized bubbles within our preparation (Fig. 3). While <0.01% by percentage, this fraction could generate a comparable sub-harmonic contrast signal owing to the strong dependence of scattering cross-section on bubble radius (~$r^6$)[26]. However, our results infusing the micron-sized MB fraction and 1–2 μm size isolated microbubbles which lack any persistent contrast signal argue against any contribution from microbubbles to the disease-dependent islet contrast signal. We also measure bubble accumulation more effectively using higher transmission frequencies which is consistent with sub-micron sizes. Conversely the initial vascular filling is measured less effectively using higher center frequencies, which is consistent with microbubbles being detected, but only during the initial vascular filling. Therefore, our results strongly indicate that the signal indicating insulitis results from sub-micron sized bubbles extravasating through the microvasculature rendered more permeable due to insulitis and local inflammation.

Studies using magnetic nanoparticles and MRI to detect inflammation rely on islet-resident macrophage uptake of the magnetic nanoparticles to retain contrast at the site of insulitis and inflammation. In this study we did not measure the kinetics for nanobubble clearance from the islet microenvironment, being >30 min. However, the kinetics of clearance from the vasculature is <10 min[40]. This rapid vascular clearance, consistent with many bubble contrast agents for ultrasound, allows the specific islet signal to be separated within the same measurement window. Nevertheless, testing whether accumulation persists over many hours or days will be a goal for future work and is highly relevant to using NBs as a vehicle for therapeutic delivery to the islet, as discussed above.

**Potential for translation of CEUS for clinical diagnosis.** Our results also suggest imaging NB accumulation within the pancreas may be a clinically deployable approach for diagnosing insulitis and pre-clinical asymptomatic T1D. There have been limited studies of human islet microvascular function compared to those in mouse. However, the use of MRI to measure magnetic nanoparticle accumulation within the pancreas has been reported in human subjects with T1D, and these studies have demonstrated clear differences between healthy patients and those recently diagnosed with T1D[18,19]. This indicates that the islet microvascular permeability is likely similarly altered in human, despite differences in the intensity and duration of insulitis. Therefore, nanobubbles will likely also accumulate in the human islet microenvironment in T1D, although the precise microvascular permeability changes are still unknown. There are also other important considerations to discuss before this approach can be clinically deployable.

Microbubbles (e.g. Lumason®, Bracco) are FDA approved for use in liver applications in adult and pediatric patients. Furthermore, they have had additional off label applications, including visualizing pancreatic tumors[41,42]. The use of ultrasound modalities provides several advantages over other clinical imaging modalities, including wide-spread deployment, cost effectiveness, high spatial and temporal resolution and the lack of ionizing radiation enabling repeated measurements. While nanobubbles are not clinically approved, they are synthesized using many similar components to clinically applied formulations. These approved formulation are polydisperse, and thus include a significant fraction of

sub-micron-sized bubbles. While the nanobubble formulation we use includes almost all bubbles within the sub-micron range required for islet accumulation, the sub-micron fraction within a polydisperse formulation is substantial and could still provide islet accumulation and contrast signal, despite the micron-sized fraction leading to inefficiencies. Thus, translating to clinical applications for diabetes detection using a polydisperse formulation is not insurmountable.

Nevertheless, the nanobubble formulation we use for imaging islet infiltration is preliminary, and will require further optimization in terms of bubble size or circulation time, to optimize islet targeting. In addition, improving the efficiency of bubble preparation to generate a higher yield of sub-micron sized nanobubbles while producing less waste products is needed. This could include optimizing the method used to drive bubble amalgamation (e.g. use of probe sonicator) to create a higher concentration of nanobubbles, or using alternative methods of bubble size reduction, such as filtration, to increase the yield and reduce waste.

We utilized a small animal ultrasound machine in these studies with frequencies greater than those clinically applied for pancreas imaging[43,44]. We did not measure any contrast elevation at 12.5 MHz, a frequency closer to those used clinically. However, larger nanobubbles that have a resonant frequency closer to clinically used frequencies, but which are still sub-micron in size may provide disease-dependent islet accumulation. Indeed nanobubble imaging has been performed using clinically relevant frequencies, including the detection of increased persistence in certain classes of tumors over other classes[39]. Examining precisely how different sized nanobubbles accumulate within the islet under differing levels of insulitis and disease progression will be critical for future applications. Nevertheless, higher frequency clinical transducers are becoming more widely available and utilized, especially on the pediatric population. Intraductal ultrasound also utilizes much higher frequencies, making the current sized nanobubbles still feasible for clinical application.

While insulitis is less aggressive in human compared to mouse, the sensitivity of our measurements to insulitis even in 4w NOD indicates this approach will be applicable during the pre-symptomatic phase in human. We were also able to distinguish NOD mice that showed slow or absent progression to diabetes from the rest of the animals. Thus, measuring nanobubble accumulation in the pancreas may provide sensitivity surpassing other approaches in detecting the progression of human diabetes at an early stage.

In summary, we present a non-invasive, deployable, ultrasound imaging modality for assessing insulitis in pre-symptomatic type1 diabetes. As a result of increased islet-vascular permeability during the progression of disease, sub-micron sized nanobubble agents accumulate within the islets. The increase in ultrasound contrast signal originating from the islets allows for diagnosis of disease progression. This may provide a convenient approach to examine human islet microvascular permeability in patients in the asymptomatic phase of T1D.

## Methods

**Animals.** All animal procedures were performed under protocols approved by the Institutional Animal Care and Use Committee of the University of Colorado Anschutz Medical campus. All procedures were in accordance with institutional guidelines and ethical regulations, under the Office of Laboratory Animal Research whose facilities are AAALAC accredited (file number 00235). Female NOD mice were purchased from Jackson Laboratories (Bar Harbor, ME) at age 4–10 weeks, and imaged at either 4 weeks or 10 weeks of age. Female Rag1ko animals were bred in house and imaged at either 4 weeks or 10 weeks of age. NOD-SCID animals were purchased from Jackson Laboratories at age 10–14 weeks, and imaged at 12–24 weeks of age. Throughout the study, animals were monitored weekly for blood glucose concentration utilizing a blood glucometer (Bayer).

**Nanobubble synthesis and characterization.** Nanobubbles (NB) were prepared as previously described with minor modifications[24,25,45]. Briefly, a lipid solution consisting of 6 mg DBPC (1,2-dibehenoyl-sn-glycero-3-phosphocholine), 1 mg DPPA (1,2-dipalmitoyl-sn-glycero-3-phosphate), 2 mg DPPE (1,2-dipalmitoyl-sn-glycero-3-phosphoethanolamine) (Avanti Polar Lipids, Pelham, AL), and 1 mg mPEG-DSPE (1,2-distearoyl-sn-glycero-3-phosphoethanolamine-N-[methoxy (polyethylene glycol)-2000] (ammonium salt)) (Laysan Lipids, Arab, AL) was prepared in 1 mL of chloroform. Chloroform was removed via evaporation overnight under vacuum. The lipid film was hydrated in a mixture of glycerol (50 μL) and 1 mL of 0.6 mg/mL Pluronic L10 in PBS, at 75 °C for 30 min. The solution was placed in a 3 mL headspace vial and air was manually exchanged with perfluoropropane ($C_3F_8$, AirGas, Cleveland, OH) gas. The lipid solution was activated by mechanical agitation for 45 s and then centrifuged upside down at 50 rcf for 5 min. NBs were then carefully isolated from the bottom of the inverted vial at a fixed distance of 5 mm. To prepare rhodamine-labeled NBs, 50 μL (1 mg/mL solution in chloroform) of Liss Rhod PE (1,2-dipalmitoyl-sn-glycero-3-phosphoethanolamine-N-(lissamine rhodamine B sulfonyl) (ammonium salt)) was added to the NB lipid mixture in chloroform and the procedure carried on as described above. Following activation, a 28.5-gauge needle was used to remove the NB fraction, which was then diluted in PBS (not degassed) by a factor of 1:5 and pipetted into a 28.5G insulin syringe for subsequent use. NBs were used immediate after activation. NBs were activated within 30 days of the synthesis of the hydrated lipid solution.

Bubbles were first characterized at Case Western via resonant mass measurement (RMM) (Archimedes, Malvern Panalytical Inc., Westborough, MA, USA) which measures particle size, size distribution, and concentration. In RMM, measurements are performed using a sensor with a microelectromechanical systems (MEMS) resonator, which contains a microfluidic channel embedded in a resonating cantilever under vacuum to detect, count, and measure the buoyant mass of the particles in the liquid passing through the channel. The RMM nanosensor was used to characterize the nanobubbles, which provides measurement between 100 nm to 2 μm. Sensors were calibrated using NIST traceable 565 nm polystyrene bead standards (ThermoFisher 4010S, Waltham MA, USA). Nanobubbles were diluted 1:100 with phosphate buffered saline (pH 7.4) and a total of 1000 particles were measured for each trial performed ($n = 3$). The sensor and microfluidic tubing were cleaned with deionized water in between each run.

Size-isolated microbubble (SIMBs) contrast agent was purchased from Advanced Microbubbles Laboratories (Boulder, CO) and infused with a 28.5 G insulin syringe. Approximately 10 million microbubbles were infused in a 100 μl solution.

**Contrast-enhanced ultrasound (CEUS) imaging.** General anesthesia was established with isoflurane inhalation for a total of 40–50 min for all animal imaged. Prior to imaging, a custom made 27 G ½″ winged infusion set (Terumo BCT, Lakewood, CO) was attached to a section of polyethylene tubing (0.61 OD × 0.28 ID; PE-10, Warner Instruments) and was inserted in the lateral tail vein and secured with VetBond (3M). Abdominal fur was removed using depilatory cream, and ultrasound coupling gel placed between the skin and transducer. Foot pad electrodes on the ultrasound machine platform monitored the animal's electrocardiogram, respiration rate, and body temperature. All animals were constantly monitored throughout the imaging session to maintain body temperature and respiration rate.

A VEVO 2100 small animal high-frequency ultrasound machine (Visual Sonics, Fujifilm, Toronto, Canada) was used for all experiments. Data collection used Vevo Lab Version 1.7.0 (Build 7071). For CEUS imaging a MS201 linear array transducer was initially used (Fig. 1b,c) at a frequency of 12.5 MHz or 18 MHz; and subsequent experiments utilized a MS250 linear array transducer at a frequency of 18 MHz. B-mode imaging (transmit power 100%) was performed prior to NB or MB infusion to identify anatomy of the pancreas body, based on striated texture and location in relation to the spleen, kidney, and stomach (Fig. 1a, Supplementary Fig. 1). Following identification of the pancreas and selection of a region of interest, sub-harmonic contrast mode was initiated. For the initial contrast mode experiments, acquisition settings were set at: transmit power 4% or 10%, frequency 12.5 MHz or 18 MHz, standard beamwidth, contrast gain of 30 dB, 2D gain of 18 dB, with an acquisition rate of 26 frames per second. For all subsequent experiments using contrast mode, acquisition settings were set at: transmit power 10%, (MI = 0.12), frequency 18 MHz, standard beamwidth, contrast gain of 30 dB, 2D gain of 18 dB, with an acquisition rate of 26 frames per second. Gating to remove movements as a result of animal breathing was carried out manually or in MATLAB (MathWorks, Natick, MA) (versions R2016b and R2018a).

Following nanobubble (NB) activation and separation (see above), NBs were injected as a single bolus of 100 μl solution of $2 \times 10^{11}$ NB/ml into the lateral tail vein via the catheter. Background data were acquired for 3 min prior to injection. Time courses were stored at 30 s intervals up to a 30 min duration post injection. Identical procedures were used with MBs. The continual imaging for >30 min using 18 MHz and 10% transmission power did not significantly impact the contrast signal generated by pancreas-accumulating NBs (Supplementary Fig. 4).

For analysis of NB contrast, regions of interest for the pancreas and kidney, were identified by the B-mode image based upon anatomical features and texture in VevoCQ Analysis;, and manually adjusted prior to analysis of the infusion to avoid

small regions of very high contrast in the background image. Data were exported into Matlab, where the NB signal was background subtracted by the averaged contrast intensity taken before injection. Each time course was normalized to its pre-injection background. Similar trends were seen if the background subtracted signal was not normalized to background, albeit increased variability (Supplementary Fig. 5).

**Isolation and adoptive transfer of diabetogenic splenocytes.** Splenocytes were isolated from diabetic female NOD mice (hyperglycemic for <1 week), manually dissociated and counted in cold HBSS (without $MgCl_2$ and $CaCl_2$). Leukocytes were counted to determine an estimate of cellular density. 12–14-week-old NOD-SCID mice received a single intraperitoneal (I.P.) dose of $20 \times 10^6$ leukocytes resuspended in HBSS. Control animals were injected with an equivalent volume of HBSS without leukocytes. A subset of NOD-Scid animals that had undergone splenocyte adoptive transfer were injected with a single dose of 20 mg anti-mouse CD4, clone GK1.5 (BP0003-1; BioXCell, W. Lebanon, NH) two weeks following splenocyte delivery.

**Histology and insulitis morphology.** For assessment of insulitis, all animals were anesthetized by I.P. injection of Ketamine (80 mg/kg) and Xylazine (16 mg/kg) until no longer reactive to toe pinch, pancreata were dissected and mice were euthanized by exsanguination and/or Bilateral thoracotomy. Pancreata were fixed in paraformaldehyde at 4°C rocking overnight and embedded in OCT blocks similar to published protocols[11]. 8 μm sections from at least three tissue depths were stained by Hematoxylin and Eosin (H&E) for evaluation of islet monocyte infiltration. Images were acquired on an Eclipse-Ti wide field microscope with a $20 \times 0.75$ NA Plan Apo objective with a color CCD camera.

Images of islets were scored based on the extent of infiltration/insulitis: grade 0, no insulitis; grade 1, peri-insulitis with immune infiltrate bordering; grade 2, immune infiltrate penetrating the islet, covering <50% of the islet area; grade 3, immune infiltrate penetrating the islet, covering >50% of the islet area. A minimum of three different tissue depths and at least 30 non-overlapping islets per animal were analyzed. Weighted averages were calculated for each animal.

For histological assessment of NB extravasation 4 w or 10w female NOD mice (JAX) or Rag1ko mice received a single bolus injection of 100 μl solution of $2 \times 10^{11}$ NB/ml of rhodamine-labeled NBs. Following contrast imaging, mice were anesthetized by I.P. injection of ketamine (80 mg/kg) and xylazine (16 mg/kg). The pancreas was dissected and fixed in 4% PFA on ice for 1 h and cryoprotected in 30% sucrose overnight or until the tissue sank. Pancreata were embedded in OCT medium, frozen in cryomolds, and cryosectioned at 8 μm sections. Sections were imaged on an LSM800 confocal microscope (Zeiss), with at 561 nm excitation using a x63 1.2NA objective and pinhole settings to provide 1 μm z-section thickness throughout the tissue depth. Separate images were taken of exocrine tissue at locations anatomically isolated from the islets. Rhodamine coverage was calculated in MATLAB as the area of rhodamine positive pixels (pixels with fluorescence intensity significantly above the background fluorescence intensity) across the islet and expressed as a fraction of total islet area.

For assessment of NB coverage within and outside of the islet vasculature, 10w female NOD mice (JAX) or Rag1ko mice received a single bolus of 100 μg of Texas Red labeled *Lycopersicon esculentum* (tomato) lectin (Vector Laboratories, #TL1176, Burlingame, CA) via tail vein injection, under ketamine (40 mg/kg) and xylazine (8 mg/kg) anesthesia. Lectin was allowed to circulate for 5 min, followed by another injection of ketamine (40 mg/kg) and xylazine (8 mg/kg) to induce deep anesthesia. Following pancreas isolation, tissues were fixed, cryoprotected, embedded, sectioned and imaged as above. Rhodamine coverage was calculated in MATLAB as above for regions that showed significant lectin staining (vasculature) and absent lectin staining (tissue).

For assessment of insulin and glucagon coverage (β-cell and α-cell coverage), cryosections were first blocked in 2% normal donkey serum (NDS) in PBS with 0.1% Triton-X (PBST) for 30 min at room temperature before incubation overnight at 4°C with guinea pig-anti-insulin primary antibody (Autostainer Link 48, IR00261-2, Agilent/DAKO, Santa Clara, CA) diluted 1:5 and rabbit-anti-glucagon primary antibody (2760S, Cell Signaling Technology, Danvers, MA) diluted 1:250. Slides were washed 4 times in PBST, and then incubated in AlexaFluor 555 goat anti-guinea pig secondary antibody (A-21435, Thermo Fisher/Invitrogen, Carlsbad, CA) diluted 1:500 and AlexaFluor 488 donkey-anti-rabbit secondary antibody (A21206, Thermo Fisher/Invitrogen, Carlsbad, CA) diluted 1:500 for 2 h at room temperature. All primary and secondary antibodies were diluted in 2% NDS in PBST. After 4 washes with PBST, slides were stained with DAPI (D1306, Thermo Fisher/Invitrogen, Carlsbad, CA) diluted 1:1000 in PBST for 15 min at room temperature, washed twice more in PBST, and mounted with Vectashield Hardset Antifade Mounting Media (H-1400, Vector Laboratories/Maravai LifeSceinces, Burlingame, CA). Sections were imaged on an LSM800 confocal microscope (Zeiss), using a x63 1.2NA objective and pinhole settings to provide 1 μm z-section thickness throughout the tissue depth. Insulin was imaged at 561 nm excitation, glucagon was imaged with 488 nm excitation, and Dapi was imaged with 405 nm excitation. Insulin and glucagon coverage was calculated in MATLAB as the area of insulin or glucagon staining positive (pixels with fluorescence intensity significantly above the background fluorescence intensity) expressed as a fraction of total pancreas area.

**Assessment of NB and MB size.** For optical quantification, the NB fraction was activated and the 100 μl NB solution was serially diluted by a factor of either $10^3$, $5 \times 10^3$ or $10^4$ with PBS and plated on glass slides. The slides were sealed with a coverslip and CoverGrip (Biotium, Fremount, CA). The plated fractions were imaged using 561 nm excitation on an LSM800 confocal microscope with a x63 1.2NA oil immersion objective (Zeiss), using a pixel size of 198 nm. Nano-sized and micron-sized objects were identified using the Analyze Particle function in ImageJ (Version: 2.0.0-rc-69/1.52i). Nano-sized objects were classified as having a diameter <1 μm, and "micron-sized" objects a diameter >1 μm. Only samples diluted by a factor of $10^4$ were used to count nano-sized objects owing to the high density preventing accurate counting under lower dilutions. Total counts were estimated based on a colony-forming unit calculation: bubbles/ml = $(N * \mathrm{df}) / (V_{pl} * A_{Im}/A_{pl})$, where $N$ is the number of objects counted of the size classification; df is the dilution factor applied; $V_{pl}$ is the volume of bubble solution that was plated; $A_{Im}$ is the area of the plated bubble solution that was imaged; and $A_{pl}$ is the total area covered by the plated bubble solution.

**Statistics and reproducibility.** Unless otherwise indicated, all error bars represent standard error in the mean (s.e.m.). Comparisons between experimental groups utilized a paired a Students $t$ test in the case of comparing two groups, or ANOVA in the case of comparing multiple groups. All representative data are accompanied by quantification indicating the number of independent repetitions.

**Reporting summary.** Further information on research design is available in the Nature Research Reporting Summary linked to this article.

## Data availability

The data generated and analyzed during the current study are available from the corresponding author on reasonable request. The source data underlying Figs. 1–5 and Supplementary Figs. 2–6 are provided as a Source Data file.

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

## Acknowledgements

Richard KP Benninger (University of Colorado) is the guarantor of this work and, as such, had full access to all the data in the study and takes responsibility for the integrity of the data and the accuracy of the data analysis. This work was supported by Juvenile Diabetes Research Foundation Grants 1-INO-2017-435-A-N, 5-CDA-2014-198-A-N; and NIH grants R01 DK102950, R01 DK106412 (to RKPB). DR has received funding from NIH training grant T32 HL072738-14; and NSF Grant HRD-1301885 (sub-award, G-8960-1). This work was also supported by funding from DOD Prostate Cancer Research Program under Award No. W81XWH-l6-l-037I and NIH grant R01EB025741 (to AAE). The funders had no role in the study design, data collection and analysis, decisions to publish, or preparation of the manuscript.

## Author contributions

D.R. designed and performed experiments, analyzed data, wrote the manuscript; C.H. and E.A. synthesized and characterized the nanobubbles; D.S.L. performed experiments; L.A.P. performed experiments and analyzed data; S.P. analyzed data; VP performed experiments and analyzed data; A.A.E. conceived of the idea, designed experiments and edited the manuscript; R.K.P.B. conceived of the idea, designed experiments and edited the manuscript.

## Competing interests

The authors declare no competing interests.
