## [Peer Review File · Nature Communications]

Reviewer #1:

Remarks to the Author:

This manuscript describes imaging of diabetes-related insulinitis in murine models, via accumulation of nanobubbles in the interstitial space of pancreas. The idea is that in response to inflammation vasculature in pancreas becomes leaky, leading to extravasation of nanoparticles that circulate through the bloodstream.

There are several issues that need to be addressed.

First is the missing details of the preparation (that are also lacking from ref. 25 and 45).

Specifically, lipid concentration information is not presented, neither in the text, nor in the supplement, so that the information is not sufficient to repeat this study. While pluronic concentration is listed, lipid concentration is presented only as mass ratio; Likewise, respectively, for the fluorescent dye amount, the ratio to other lipid components is not clear.

It is not clear which gas is inside the nanobubbles, and in what quantity. Presumably it is perfluoropropane? How much of perfluorocarbon was associated with nanobubble fraction? Dilution with PBS is mentioned in the description: was that PBS degassed? If not, there is a risk that dissolved air, which is present in aqueous medium, could have interacted with and exchanged into the nanobubble preparation.

In this manuscript, nanobubble formulation is compared with Lumason/Sonovue microbubbles, a commercial formulation. Another commercial formulation, Definity, might be a much more relevant comparison, because it has ~90% of the bubble number concentration as nanobubbles.

Comparison with Definity in this animal study will be able to address directly the issue of importance of pluronic in the nanobubble formulation (Definity lacks pluronic). It may also be of highest practical importance, because if nanobubbles of the clinical Definity preparation extravasate in this model, potential of clinical use of the proposed approach may become much closer.

The use of ultrasound at 12.5MHz or 18 MHz, and 4% or 10% transmit power, may require a more detailed explanation. It is not clear if power selection is used to minimize microbubble destruction, or a power level above certain threshold, and at a particular frequency, would be necessary to obtain acoustic backscatter signal from nanobubbles. Would intermittent imaging at higher power levels, and not continuous imaging at 26 fps, for 30 min, as listed here, be advantageous? Would 18 MHz be obligatory? The reason for that question is the level of applicability of the proposed approach for clinical use: it is unlikely that penetration of ultrasound at that frequency would be sufficient to image human pancreas, which is located away from the skin, and represents much denser tissue than in a mouse. This needs to be clarified.

Reviewer #2:

Remarks to the Author:

Ramirez et al showed that contrast-enhanced ultrasound with sub-micron sized contrast agents can detect insulinitis in 8 and even 4 week old type 1 diabetic NOD mice as well as in NOD-SCID mice given autoreactive T cells. This observation is very interesting and has the potential to develop a non-invasive and deployable ultrasound imaging for assessing insulinitis in pre-symptomatic type 1 diabetes. However, there are major concerns about the sensitivity and reliability of the technique.

1. The title "detects insulinitis and beta cell mass decline..." is an overstatement. There is no evidence that the described technique can detect "beta cell mass decline".

2. Although the data showed that the technique can detect insulinitis in 8 and even 4 weeks old NOD mice, there were only 3 mice of 4 weeks of age. It is true that NOD mice start to have insulinitis at 4 weeks of age, but it is very heterogeneous. Some mice have insulinitis and some do not. In order to prove the reliability and sensitivity of the technique, many more 3 and 4 weeks old mice need to be checked and the percentage of mice with or without insulinitis in each age group need to be calculated. The insulinitis or no insulinitis detected by the ultrasound technique need to be validate by histopathology studies with HE staining of pancreatic tissues.

Reviewer #3:

Remarks to the Author:

Dr. Ramirez et al present work in which they propose contrast-enhanced ultrasound using nanobubble contrast agents to detect microvascular permeability as a measure of insulinitis and beta cell mass in pre-clinical type 1 diabetes. They show analyses demonstrating that contrast uptake in pancreas is increased in NOD mice compared to Rag1ko mice at very early and later stage timepoints and increased with age in NOD mice. Importantly, the authors also demonstrate that NOD-scid mice receiving adoptive transfer of splenocytes develop an increased signal, reflective of a direct relationship between increased signal and islet autoimmunity. The manuscript presents a novel and timely idea. This modality, if able to accurately identify preclinical diabetes in humans, would have clear clinical use, as prediction of T1D is challenging, and earlier prediction could improve results of therapeutics aimed at delaying or preventing disease. However, concerns regarding completeness of the islet phenotyping, the true ability of the imaging to differentiate between more subtle differences in insulinitis, and whether the enhanced uptake is actually measuring insulinitis limits enthusiasm for the work. Especially given differences in insulinitis in humans and NOD mice, there is significant concern that a modality that is able to discriminate between the presence and absence of insulinitis in these mice vs immunodeficient mice will be sensitive enough to detect differences among humans at risk for T1D.

Major

1. Figure 2g- why wasn't insulin staining used to identify islets? It seems important to know what types of islet cells the rhodamine contrast accumulates in. Also in this figure – the rhodamine staining does not appear visible within the islet for the NOD mouse image example. It is surprising that this was the best image given the pronounced difference observed in the quantification.
2. Figure 2J- the correlation of signal with insulinitis seems to be driven by differences between the rag1ko and NOD strains, rather than a strong correlation within the mice with insulinitis (except for the one NOD outlier with a very increased signal). This correlation analysis should just be performed with NOD mice and if the data are not normally distributed, should be analyzed with a nonparametric statistical test.
3. If the uptake of these NBs are due to insulinitis, it is surprising and confusing that a relationship with insulinitis severity would be present on a whole pancreas level, but not at the level of the individual islet. The authors speculate this may be due to macrophage recruitment in unaffected islets. Was this the case in histologic analyses?
4. The title of the paper and the abstract refer to the ultrasound detecting decline of beta cell mass but no studies of beta cell mass are shown. What were relationships of contrast uptake with beta cell mass?
5. The authors propose this modality as a measure to predict future type 1 diabetes in autoantibody positive individuals since not all individuals with islet autoantibodies develop t1d. Presumably, these studies would not be applied to population-based screening. Thus, NOD mice who do not develop diabetes seem to be a more appropriate control than Rag1ko mice, which exhibit very large differences in uptake. Among NOD mice, is an increase in signal predictive of diabetes development?
6. The ability of this technology to differentiate more subtle differences in insulinitis (ie nonprogressor vs progressor NOD mouse) is especially important in the context of translating this technology to human T1D. As noted by the authors in the discussion, insulinitis is typically less

aggressive in human T1d than in the NOD mouse, with differences in the duration and intensity. The authors address this in the discussion by saying that other studies using other nanoparticle modalities in humans have identified altered microvascular permeability in humans with T1D, despite differences in insulinitis, which seems to discredit the key theme of the rest of the paper that the permeability is directly related to insulinitis.

Minor

1. 2j- the islet does not appear to be circle as indicated in the figure legend
2. For all immune fluorescent staining, would be helpful to label figure with antigens represented by different colors (eg 2j)

Response to Reviewers.

We thank the editor and reviewers for their efforts. Below follows our detailed response to each of the reviewers' comments (reproduced in *italics*), together with a description of where and how we have modified the manuscript. Where relevant we include data from new experiments. All changes in the manuscript are indicated in **red** and we underline the position in the manuscript we make relevant changes.

Reviewer #1 (Remarks to the Author):

This manuscript describes imaging of diabetes-related insulinitis in murine models, via accumulation of nanobubbles in the interstitial space of pancreas. The idea is that in response to inflammation vasculature in pancreas becomes leaky, leading to extravasation of nanoparticles that circulate through the bloodstream. There are several issues that need to be addressed.

We thank the reviewer for raising the below points which we agree with and discuss further below

First is the missing details of the preparation (that are also lacking from ref. 25 and 45). Specifically, lipid concentration information is not presented, neither in the text, nor in the supplement, so that the information is not sufficient to repeat this study. While pluronic concentration is listed, lipid concentration is presented only as mass ratio; Likewise, respectively, for the fluorescent dye amount, the ratio to other lipid components is not clear.

We thank the reviewer for pointing out these missing details (and those below) that would prevent reproduction of the experiments. We now include details for lipid concentrations used (6mg/ml DBPC, 1mg/ml DPPA, 2mg/ml DPPE, 1mg/ml mPEG-DSPE) and fluorescent dye quantity 50ul or a 1mg/ml solution added to 1ml lipid solution). See Methods page 15 paragraph 2 (lines 493-508). .

It is not clear which gas is inside the nanobubbles, and in what quantity. Presumably it is perfluoropropane? How much of perfluorocarbon was associated with nanobubble fraction? Dilution with PBS is mentioned in the description: was that PBS degassed? If not, there is a risk that dissolved air, which is present in aqueous medium, could have interacted with and exchanged into the nanobubble preparation.

Perfluoropropane was used. Based on Archimedes measurements we estimate nanobubbles contain 5.95nL/ml gas. The PBS was not degassed and we appreciate a small amount of air exchange could occur, as could also occur with the bubbles in blood and extra-cellular fluid following infusion. We also state this. See Methods page 15 paragraph 2 (line 502 and line 510).

In this manuscript, nanobubble formulation is compared with Lumason/Sonovue microbubbles, a commercial formulation. Another commercial formulation, Definity, might be a much more relevant comparison, because it has ~90% of the bubble number concentration as nanobubbles. Comparison with Definity in this animal study will be able to address directly the issue of importance of pluronic in the nanobubble formulation (Definity lacks pluronic). It may also be of highest practical importance, because if nanobubbles of the clinical Definity preparation extravasate in this model, potential of clinical use of the proposed approach may become much closer.

In the manuscript we actually compared the nanobubble formulation with size-isolated microbubbles from Advanced Microbubble Labs, showing lack of extravasation and accumulation for 3-4µm size ranges and 1-2µm size ranges. We appreciate the value of making the comparison with DEFINITY bubbles, given the sub-micron component that lacks Pluronic. We are unable to obtain DEFINITY (Lantheus) bubbles, however we synthesized a formulation of polydisperse bubbles lacking pluronic and including the main components of DEFINITY (methods below). As one can see (data below) the signal elevation in the pancreas is smaller than that of nanobubbles. However, it is significantly greater than zero. Further, this small elevation was missing in measurements of the kidney where no signal elevation was observed for

'DEFINITY-like bubbles'. This data therefore indicates that 'DEFINITY-like bubbles' do show disease-dependent extravasation in the pancreas of models of T1D, but are less effective (or in a smaller quantity) than nanobubble formulation we utilized. We think this data indicates a more substantial study is needed, using the approaches and models presented in this manuscript as part of a dedicated study looking at DEFINITY predicting T1D progression. As such we do not include this data within the revised manuscript.

Definity-like bubbles were prepared based on a previously published patent* and was prepared by completely dissolving 0.4 mg DPPC (1,2-dipalmitoyl-sn-glycero-3-phosphocholine), 0.045 mg DPPA, and 0.3 mg mPEG-DPPE (1,2-dipalmitoyl-sn-glycero-3-phosphoethanolamine-N-[methoxy(polyethylene glycol)-5000] (ammonium salt)) in 0.1 mL of propylene glycol at 80 °C (water bath).⁴ A pre-heated mixture of 0.1 mL of glycerol and 0.8 mL PBS at 80 °C was then added to the lipid solution dissolved in propylene glycol. The lipid solution was then transferred to a 3 mL headspace vial, capped and sealed. Gas exchange and bubble activation followed the same procedure as described above.

* Unger, E. C. & Evans, D. C. Patent: WO2015192093A1. Phospholipid composition and microbubbles and emulsions formed using same. 53 (2015).

Mean contrast elevation (Signal-Background/Background) averaged over 20-25minutes after infusion for NOD mouse (n=4). Left: Comparison of contrast elevation for DEFINITY-like bubbles and nanobubble formulation. Right: Comparison of contrast elevation for DEFINITY-like bubbles in pancreas and kidney of NOD mouse.

The use of ultrasound at 12.5MHz or 18 MHz, and 4% or 10% transmit power, may require a more detailed explanation. It is not clear if power selection is used to minimize microbubble destruction, or a power level above certain threshold, and at a particular frequency, would be necessary to obtain acoustic backscatter signal from nanobubbles. Would intermittent imaging at higher power levels, and not continuous imaging at 26 fps, for 30 min, as listed here, be advantageous? Would 18 MHz be obligatory? The reason for that question is the level of applicability of the proposed approach for clinical use: it is unlikely that penetration of ultrasound at that frequency would be sufficient to image human pancreas, which is located away from the skin, and represents much denser tissue than in a mouse. This needs to be clarified.

We vary the transducer frequency (12.5MHz and 18MHz) given different size bubbles will be resonant at different frequencies, hence 12.5MHz may preferentially detect larger bubbles and 18MHz may preferentially detect the smaller bubbles that we hypothesized would more likely extravasate.

We vary the transmit powers since we did not initially observe a contrast elevation using 4% but did with 10%. Thus, to address the reviewer's initial question – it appears a power level above some threshold is required to obtain sufficient backscatter.

We performed an additional experiment where we elevated the transmit power and imaged for a very short window (~10s) every 5 minutes, similar to the previous supplemental figure S4 (now figure S2). In this case increasing the power intermittently for the 12.5MHz frequency does not lead to any contrast elevation and thus does not detect the nanobubble extravasation. We appreciate the reviewer's point regarding clinical frequencies, but do note a few factors: a) Intraductal ultrasound would be possible with this frequency, which is routinely used clinically b) As indicated in the size distribution, the diameter for these nanobubbles at ~200nm is small. While we know 1-2µm diameter is too large for extravasation, there likely exists a bubble size (e.g. ~500nm diameter) that will be detected by a 12.5MHz transducer or lower

frequency compatible with imaging the human pancreas, but which is also sufficient for non-linear scattering with these bubble sizes. This requires further investigation to design and test the acoustic properties of specific sub-micron size ranges.

As such we edit the results section that presented these data to describe the rationale for multiple frequencies and transmit powers – see Results page 5 paragraphs 1 (line 127-128) and 2 (line 135-142). We also discuss these results and their relevance for clinical imaging - see Discussion page 13 paragraph 4 (lines 454-462).

Mean contrast elevation (Signal-Background/Background) for high transmit powers for differing frequencies. Left: Time-course of contrast elevation with initial intermittent imaging followed by continual imaging for 10w female NOD mice (n=4). Right contrast elevation averaged over 40-45minutes after infusion.

Reviewer #2 (Remarks to the Author):

Ramirez et al showed that contrast-enhanced ultrasound with sub-micron sized contrast agents can detect insulinitis in 8 and even 4 week old type 1 diabetic NOD mice as well as in NOD-SCID mice given autoreactive T cells. This observation is very interesting and has the potential to develop a non-invasive and deployable ultrasound imaging for assessing insulinitis in pre-symptomatic type 1 diabetes. However, there are major concerns about the sensitivity and reliability of the technique.

We are pleased to hear the reviewer finds the results we present to be very interesting. We thank them for their helpful comments which we now address.

1. The title “detects insulinitis and beta cell mass decline...” is an overstatement. There is no evidence that the described technique can detect “beta cell mass decline”.

We agree with this statement. We have since measured changes in beta cell mass (more specifically the mean insulin to glucagon ratio to factor out pancreas size) for NOD mice and NOD;Rag1ko mice and compared these measurements with nanobubble accumulation (rhodamine labelling). We do see a decrease in insulin coverage in NOD mice compared to Rag1ko mice, correlating with the increased NB accumulation. However, the trend for decreased insulin coverage with increased NB accumulation was not statistically significant. We appreciate this is only a correlatory measurement to detect beta cell mass, but nevertheless may be useful. Nevertheless, we discuss that measuring nanobubble extravasation is likely less predictive of beta cell mass changes/differences compared to insulinitis changes/differences. As such we have adjusted the manuscript title. We also now describe and discuss these findings – see Results page 7 paragraph 4 (lines 221-226); Discussion page 10 paragraph 3 (lines 337-342); and Methods page 18 paragraph 3 (lines 618-639). We also include glucagon coverage data further below that we do not include within the manuscript.

Insulin and glucagon staining. Left: representative images of islet from NOD mouse and Rag1ko mouse. Middle: mean insulin coverage averaged over NOD and Rag1ko pancreas. Right: correlation of NB accumulation with insulin coverage.

Left: mean glucagon coverage averaged over NOD and Rag1ko pancreas. Right: correlation of NB accumulation with glucagon coverage.

2. Although the data showed that the technique can detect insulinitis in 8 and even 4 weeks old NOD mice, there were only 3 mice of 4 weeks of age. It is true that NOD mice start to have insulinitis at 4 weeks of age, but it is very heterogeneous. Some mice have insulinitis and some do not. In order to prove the reliability and sensitivity of the technique, many more 3 and 4 weeks old mice need to be checked and the percentage of mice with or without insulinitis in each age group need to be calculated. The insulinitis or no insulinitis detected by the ultrasound technique need to be validate by histopathology studies with HE staining of pancreatic tissues.

We do acknowledge that measurements of rhodamine-labelled nanobubble extravasation and accumulation was only measured across 3 NOD and 3 Rag1ko mice. We have now doubled our group size to 6 NOD and 6 Rag1ko mice, and still observe nanobubble accumulation within the islets of 4w NOD mice and negligible levels on Rag1ko mice. We also measured insulinitis in each of these 6 NOD mice, and observe substantially lower insulinitis than at 10w, but still significant levels (mostly peri-insulinitis). See Figure 4; Results page 8 paragraph 3 (lines 274-278).

Histological analysis of 4w NOD and Rag1ko mice. Left: mean proportion of islet and exocrine tissue showing NB- rhodamine coverage from 4w NOD and Rag1ko mice. Right: mean insulinitis score averaged over pancreata of 4w NOD and Rag1ko mice. Insulin and glucagon staining. Left: representative images of islet from NOD mouse and Rag1ko mouse. Middle: mean insulin coverage averaged over NOD and Rag1ko pancreas. Right: correlation of NB accumulation with insulin coverage.

Reviewer #3 (Remarks to the Author):

Dr. Ramirez et al present work in which they propose contrast-enhanced ultrasound using nanobubble contrast agents to detect microvascular permeability as a measure of insulinitis and beta cell mass in pre-clinical type 1 diabetes. They show analyses demonstrating that contrast uptake in pancreas is increased in NOD mice compared to Rag1ko mice at very early and later stage timepoints and increased with age in NOD mice. Importantly, the authors also demonstrate that NOD-scid mice receiving adoptive transfer of splenocytes develop an increased signal, reflective of a direct relationship between increased signal and islet autoimmunity. The manuscript presents a novel and timely idea. This modality, if able to accurately identify preclinical diabetes in humans, would have clear clinical use, as prediction of T1D is challenging, and earlier prediction could improve results of therapeutics aimed at delaying or preventing disease. However, concerns regarding completeness of the islet phenotyping, the true ability of the imaging to differentiate between more subtle differences in insulinitis, and whether the enhanced uptake is actually measuring insulinitis limits enthusiasm for the work. Especially given differences in insulinitis in humans and NOD mice, there is significant concern that a modality that is able to discriminate between the presence and absence of insulinitis in these mice vs immunodeficient mice will be sensitive enough to detect differences among humans at risk for T1D.

We appreciate the enthusiastic comments the reviewer makes in stating how the manuscript presents a novel and timely idea. We understand the concerns the reviewer raises and have addressed them as detailed below.

Major

1. Figure 2g- why wasn't insulin staining used to identify islets? It seems important to know what types of islet cells the rhodamine contrast accumulates in. Also in this figure – the rhodamine staining does not appear visible within the islet for the NOD mouse image example. It is surprising that this was the best image given the pronounced difference observed in the quantification.

We have performed insulin and glucagon staining to compare changes in beta cell mass with our measurements (see response to reviewer 2). However, we did not do this in conjunction with rhodamine assessment, to avoid washing away any rhodamine accumulation during sample staining.

We do not think that the nanobubbles and thus fluorophore are specifically targeting beta cells, rather they accumulate more generally at the site of immune infiltration which is localized across the islet and in the islet periphery. This localization is actually similar to prior reported studies that examined a magnetic nanoparticle accumulation in infiltrated islets – Denis et al PNAS (2004) p12634 (<https://www.pnas.org/content/101/34/12634.long>). Also, we did not try and provide 'the best image' but rather an image that was more representative of what we generally observe.

In response to this comment we have included data for insulin and glucagon staining – see Figure 2m-o; Results page 7 paragraph 4 (lines 221-226); and data presented above for reviewer 2. We also discuss that nanobubble targeting is general to the site of islet infiltration and not specific to cells – see Discussion page 12 paragraph 3 (lines 405-406).

2. Figure 2J- the correlation of signal with insulinitis seems to be driven by differences between the rag1ko and NOD strains, rather than a strong correlation within the mice with insulinitis (except for the one NOD outlier with a very increased signal). This correlation analysis should just be performed with NOD mice and if the data are not normally distributed, should be analyzed with a nonparametric statistical test.

We apologize for this misleading graph. We only performed the correlation over the NOD mouse data points which shows a significant correlation. The Rag1ko data was there as a comparison to indicate negligible insulinitis – if it was included in correlation analysis it would indeed be driving the trend. As such we have removed the Rag1ko data from this graph and include a separate graph presenting just the results of insulinitis. See Figure 2 panels i-k.

3. If the uptake of these NBs are due to insulinitis, it is surprising and confusing that a relationship with insulinitis severity would be present on a whole pancreas level, but not at the level of the individual islet. The authors speculate this may be due to macrophage recruitment in unaffected islets. Was this the case in histologic analyses?

That nanobubbles are accumulating in apparently non-infiltrated islets was unexpected and surprising. However, we do not fully know the mechanistic basis by which islet microvasculature is more leaky in models of type 1 diabetes (and human type1 diabetes). We do note in prior experiments of ours we examined the link between islet microvascular dilation (which results from greater blood flow directed to the islets that is caused by islet infiltration) and T-cell infiltration, and also did not see a correlation between islet vascular morphology and infiltration. However, we did observe a correlation between islet vascular morphology and NOD mouse age (a surrogate for overall disease progression). Thus, we are confident in this data, but an explanation remains to be determined, but could be important. We speculated early macrophage recruitment, but also note other factors such as early IFNalpha release may also cause this. We appreciate this comment may be overly speculative given lack of evidence, and thus remove it – See Discussion page 11 paragraph 2 (lines 368-370).

Left: Representative images of rhodamine-lectin labelled microvasculature (to calculate diameter) and labelled infiltrating T-cells (to measure insulinitis) collected via intra-vital imaging. Right: Correlations (with 95% CI) of mean vascular diameter in islet with volume of islet infiltrated ($p=0.35$) and NOD mouse age ($p=0.046$). A significant correlation only occurs with NOD mouse age (measuring pancreas-wide disease progression).

4. The title of the paper and the abstract refer to the ultrasound detecting decline of beta cell mass but no studies of beta cell mass are shown. What were relationships of contrast uptake with beta cell mass?

As described above in response to comment 1 and reviewer 2, we have performed insulin and glucagon staining. We observed that insulin positive area was reduced in islets of NOD mice compared to Rag1ko mice, correlating with the increased nanobubble accumulation. We also observed a trend among mice for decreased insulin area for islets showing increased nanobubble and rhodamine accumulation, although this was not statistically significant.

In response to this comment we include this insulin staining data - see Figure 2m-o and Results page 7 paragraph 4 (lines 221-226). We also discuss how the level of nanobubble accumulation is a better predictor of insulinitis over beta cell mass decline - see Discussion page 10 paragraph 3 (lines 337-342).

5. The authors propose this modality as a measure to predict future type 1 diabetes in autoantibody positive individuals since not all individuals with islet autoantibodies develop t1d. Presumably, these studies would not be applied to population-based screening. Thus, NOD mice who do not develop diabetes seem to be a more appropriate control than Rag1ko mice, which exhibit very large differences in uptake. Among NOD mice, is an increase in signal predictive of diabetes development?

We propose this approach as a way to detect whether AA+ individuals will progress to diabetes and whether asymptomatic individuals receiving therapeutic treatment no longer develop diabetes.

We have examined if the contrast elevation correlates with the time to disease development in NOD mice. We identified a threshold that optimally separates NOD and Rag1ko mice. Using this threshold, we

separated mice into a group classed disease positive that showed high contrast elevation (thus high nanobubble extravasation) and a group classed disease negative that showed low or no contrast elevation. We then observed whether there was a significant difference in disease progression of these two groups of animals. NOD mice with low contrast elevation developed diabetes at a significantly later time than those with high contrast elevation. It appears that effective prediction of disease lays in identifying those mice that develop diabetes very late, which show low contrast elevations. This makes sense given we have a very sensitive measure for diabetes, yet one able to differentiate subjects.

Kaplan-Meier curve showing the progression to diabetes of NOD mice measured to have low contrast elevation following NB infusion (Disease-) or high contrast elevation (Disease+).

We also performed an additional set of experiments in the adoptive transfer group where we treated the AT mice that receive splenocytes with antiCD4 to deplete CD4+ T-cells and delay diabetes onset. This treatment led to a significant decrease in contrast elevation compared to adoptive transfer mice not receiving a treatment. Thus, we can detect the impact of a therapeutic intervention. Furthermore, those mice that showed a greater delay in diabetes development (>8weeks) did not show any significant contrast elevation measured at 4 weeks.

To address this point we include this additional data and analysis – see Figures S4 and 5e-g; Results page 6 paragraph 2 (lines 163-180) and page 9 paragraph 3 (lines 294-304). We also discuss this ability to predict diabetes progression and measure disease reversal – see Discussion page 11 paragraph 1 (lines 355-359) and page 14 paragraph 2 (lines 368-369)

(E) Kaplan Meier curve indicating proportion of mice remaining non-diabetic for AT mice and AT mice treated with antiCD4 at 2 weeks after transfer. (F) Mean time-course of sub-harmonic contrast signal in the pancreas of female NOD-scid mice 4w post splenocyte transfer (AT) and 4w post transfer where antiCD4 is provided at 2w (left), together with the mean contrast signal between 20-25 minutes (right). (G) Mean contrast signal between 20-25 minutes for AT mice treated with antiCD4 that develop diabetes >8 weeks after splenocyte transfer ('Responder') or ≤8 weeks ('Non-Responder').

6. The ability of this technology to differentiate more subtle differences in insulinitis (ie nonprogressor vs progressor NOD mouse) is especially important in the context of translating this technology to human T1D. As noted by the authors in the discussion, insulinitis is typically less aggressive in human T1d than in the NOD mouse, with differences in the duration and intensity. The authors address this in the discussion by

saying that other studies using other nanoparticle modalities in humans have identified altered microvascular permeability in humans with T1D, despite differences in insulinitis, which seems to discredit the key theme of the rest of the paper that the permeability is directly related to insulinitis.

We agree that ultimately for clinical application differentiating heavy insulinitis from no insulinitis will not be sufficient, and differentiating conditions that lead to diabetes onset from those that do not will be more important. As discussed above we can predict rapid or slow progression to diabetes, and the impact of disease delay. The ultrasound-based disease prediction measures best separate those animals that are substantially delayed from diabetes, and do not differentiate so well those animals that show rapid diabetes progression. Thus, the increased sensitivity for 'mild disease' by ultrasound measurements appears to be more compatible for human disease. We note that NOD mice are genetically identical and thus provide a very homogenous group to differentiate, as compared to humans. Thus, our being able to differentiate NOD mice based on their disease could well present a much higher bar than in human subjects.

It is important to note that it is unclear from our study here and for other studies what mechanistically links disease progression and endothelial cell permeability. We do not claim the increased permeability is directly a result of insulinitis, since islets with little insulinitis do still show nanobubble accumulation. However, it clearly is linked to overall disease progression, based upon rhodamine correlation with pancreas-wide insulinitis, and based on disease prediction and disease reversal assessments. Thus, we reword relevant parts to avoid claiming the extravasation is directly a result of insulinitis.

As such we modify the discussion to indicate that the ability to sensitively detect disease and to separate slow/none progressors to disease will be helpful in the human context – see Discussion page 14 paragraph 1 (lines 468-470). We also modify the discussion to avoid indicating that our measurement is directly a result of insulinitis – see Discussion page 10 paragraph 3 (lines 337-342) and page 11 paragraph 2 (lines 368-370).

Minor

- 1. 2j- the islet does not appear to be circle as indicated in the figure legend*
- 2. For all immune fluorescent staining, would be helpful to label figure with antigens represented by different colors (eg 2j)*

We have included the circle to indicate the islet. We also label the figure for all fluorescence images, both those originally included and new figures – see Figures 2c,m; 4c

Reviewers' Comments:

Reviewer #1:

Remarks to the Author:

This reviewer is thankful to authors for addressing almost all of the prior concerns.

There is one minor issue, which is still important to address.

In the response letter, it is stated that "we estimate nanobubbles contain 5.95nL/ml gas". This amount looks rather low. For comparison, Definity bubbles, according to FDA-approved prescribing information sheet, have ~150 microliters of gas per 1 ml of liquid, over four orders of magnitude difference, which is achieved with about an order of magnitude lower dose of lipid material than for Definity formulation. This creates an impression that most of the bubbles of the current nanobubble formulation scheme initially are of the undesirable size and have to be removed. Thus, most of the gas and lipid material is wasted. It is highly desirable to have a more efficient preparation scheme, where (as is the case for Definity) most of the bubbles generated are of sub-micrometer size, i.e., of the diameter close to what is preferred in this study. The bare minimum is to openly discuss this situation and assess the amounts of discarded lipid, surfactant, and gas, and compare that with the amounts that is within the nanobubble formulation.

Reviewer #2:

Remarks to the Author:

Previous concerns have been well addressed. No further comments.

Reviewer #3:

Remarks to the Author:

The authors included reasonable responses and new experiments to address my major concerns.

Response to Reviewers.

We thank the editor and reviewers for their efforts. Below follows our detailed response to the remaining reviewer comment (reproduced in *italics*), together with a description of where and how we have modified the manuscript.

Reviewer #1:

In the response letter, it is stated that "we estimate nanobubbles contain 5.95nL/ml gas". This amount looks rather low. For comparison, Definity bubbles, according to FDA-approved prescribing information sheet, have ~150 microliters of gas per 1 ml of liquid, over four orders of magnitude difference, which is achieved with about an order of magnitude lower dose of lipid material than for Definity formulation. This creates an impression that most of the bubbles of the current nanobubble formulation scheme initially are of the undesirable size and have to be removed. Thus, most of the gas and lipid material is wasted. It is highly desirable to have a more efficient preparation scheme, where (as is the case for Definity) most of the bubbles generated are of sub-micrometer size, i.e., of the diameter close to what is preferred in this study. The bare minimum is to openly discuss this situation and assess the amounts of discarded lipid, surfactant, and gas, and compare that with the amounts that is within the nanobubble formulation.

We would like to thank the reviewer for this comment. Indeed, as the data showed in our prior response, a fraction of the 'Definity-like' bubbles that were injected into NOD mice appeared to accumulate in the pancreas. This accumulation resulted in signal that was approximately 25% of the measured nanobubble signal. Thus following further validation the use of 'Definity-like' bubbles for this application could certainly be a potential avenue for clinical translation. However, additional in depth examination of this claim is needed to proceed along this path.

We would like to specifically comment on the gas volume. The preparation of nanobubbles indeed results in discarding approximately 50% of the total materials used in their formulation. However, the isolation of nanobubbles in the size range of 100-400 nm was crucial for demonstration of this concept. An advantage of the nanobubbles for future applications (and potential clinical uses) is that they do not need to be isolated from the total population. As reported, Definity bubbles with a reported mean diameter of 2.2 μm have a theoretical gas volume of 5.54 μm^3 per particle while the nanobubbles used in this study with a mean diameter of 0.199 μm (or 199 nm) have a theoretical gas volume of 0.0038 μm^3 per particle. This difference results in Definity bubbles having approximately 3 orders of magnitude larger volume than the nanobubbles based on size difference alone. However, the actual concentration of particles per injected volume is much higher with the NBs than with Definity. Injection of high concentration of Definity results in an injection of relatively high number of microbubbles, which in turn, leads to acoustic shadowing and prevent clear imaging of the pancreas. In addition, these microbubbles will not target the pancreas to reveal disease. Thus, there is an advantage to using a nanobubble formulation (or removing microbubbles from Definity), which do not attenuate ultrasound signal at higher concentration. There are two things we can do in the future to improve the protocol to generate a higher yield of nanobubbles and produce less waste: (1) Optimize the method used to drive bubble amalgamation/self-assembly (e.g. use of probe sonicator) to create a higher concentration of nanobubbles with less waste, and (2) use alternative methods of bubble size reduction, such as filtration, to increase the yield and reduce waste.

In response to this comment we elaborate on these points within the discussion (page 13 lines 725-730)